# An integrated transcriptomic and proteomic map of the mouse hippocampus at synaptic resolution

Eva Kaulich [1,4], Quinn Waselenchuk [1,2,4], Nicole Fürst[1], Kristina Desch [1], Janus Mosbacher[1], Elena Ciirdaeva[1], Marcel Juengling [1,3], Roshni Ray[1], Belquis Nassim-Assir[1], Georgi Tushev [1], Julian D. Langer [1,2] & Erin M. Schuman [1,2,3] ✉

Understanding the brain's molecular diversity requires spatially resolved maps of transcripts and proteins across regions and compartments. Here, we performed deep spatial molecular profiling of the mouse hippocampus, combining microdissection of 3 subregions and 4 strata with fluorescence-activated synaptosome sorting, transcriptomics, and proteomics. This approach revealed thousands of locally enriched molecules spanning diverse receptor, channel, metabolic, and adhesion families. Integration of transcriptome and proteome data highlighted proteins tightly linked to or decoupled from mRNA availability, in part due to protein half-life differences. Incorporation of translatome data identified roles for protein trafficking versus local translation in establishing compartmental organization of pyramidal neurons, with distal dendrites showing increased reliance on local protein synthesis. Classification of CA1 synapses revealed contributions from kinases, cytoskeletal elements, and adhesion molecules in defining synaptic specificity. Together, this study provides a molecular atlas of the hippocampus and its synapses (syndive.org), and offers insights into spatial transcript-protein relationships.

Transcriptomic and proteomic profiling of brain regions, neuronal populations and synapse types has revealed the brain's immense capacity for cellular and subcellular molecular diversity[1–7]. However, comprehensive studies addressing both neuronal transcript and protein diversity in parallel are lacking. The need to quantify mRNA and protein together is underscored by the consensus that abundance of either molecule is not reliably predicted by the other[8–14]. Dynamic mRNA-protein relationships are influenced by factors like mRNA stability, localization, trafficking, translation, and degradation, ultimately leading to discrepancies in their levels[14–20]. For example, during fly embryogenesis, mRNAs concentrated in specific regions influence the

local protein pool without reflecting overall mRNA abundance[9]. Similar context-dependent relationships are observed in retinal cone development in *Xenopus*[21] and in the spatio-temporal regulation of dendritic mRNA translation during neuronal plasticity, where local mRNAs are translated in response to environmental cues and protein abundances vary accordingly[22]. These factors emphasize the need for combined analyses to understand the molecular blueprints of cellular and subcellular characteristics. Still, such studies remain largely absent.

We set out to systematically characterize the transcripts and proteins defining different neuronal populations, compartments, and synapses in the hippocampus, a region with well-studied physiology,

[1]Max Planck Institute for Brain Research, Frankfurt, Germany. [2]IMPRS on Cellular Biophysics, Frankfurt, Germany. [3]Donders Institute for Brain, Cognition and Behaviour, Radboud University, Nijmegen, The Netherlands. [4]These authors contributed equally: Eva Kaulich, Quinn Waselenchuk. ✉e-mail: erin.schuman@brain.mpg.de

cellular composition, connectivity, and function. Molecular gradients and functional differences across the hippocampus[7] indicate compartmentalization of these highly organized subregions: the DG (dentate gyrus, with granule cells) and CA1-3 of Ammon's horn (*Cornu Ammonis*, pyramidal cells). Within area CA1, the excitatory pyramidal neurons can further be divided into cellular compartments as they run orthogonal to four strata: *Stratum oriens* (SO), containing basal dendrites, *Stratum pyramidale* (SP), comprised of cell bodies, and *Stratum radiatum* (SR) and *Stratum lacunosum-moleculare* (SLM), containing proximal and distal apical dendrites, respectively[23,24]. The excitatory synaptic and modulatory inputs to each layer are also distinct, including inputs from hippocampal areas CA2 and CA3, entorhinal cortex, the nucleus reunions of the thalamus, and other cortical and subcortical areas[25]. Each of the strata also houses a unique combination of interneuron cell bodies and processes[26]. The distinctive nature of their connections as well as differences in the local excitatory-inhibitory microcircuits then gives rise to different functional and plasticity properties of the strata synaptic populations[24,27,28] (SO[29–34], SP[24], SR[35–38], SLM[39–41]). While the differential distribution of some mRNAs and proteins within the hippocampus and area CA1 has been studied[15,18,19,42–46], we lack a complete picture of their spatial, cellular and subcellular relationship across these neurons and their compartments, and how this interplay contributes to functional diversity.

Here, we used parallel RNA-seq and LC-MS/MS combined with precision microdissection and Fluorescence-Activated Synaptosome Sorting (FASS[2,5,47]), a method which uses fluorescent labeling of a synaptic protein from transgenic mice to sort synaptosomes to high purity[5], to build a comprehensive map of transcripts and proteins across subregions, neuronal compartments, and synapses in the hippocampus. We achieved deep coverage of local transcriptomes and proteomes of dissected tissue, identifying thousands of molecules with targeted spatial enrichment, and used integration of both datasets to gain insights into how factors such as protein-half-lives and translation localization may contribute to this organization. From purified synaptosomes, we further outlined classes of molecules that define synaptic specificity within one brain region. This resource serves as a foundational dataset on the molecular diversity of the hippocampus, and provides in-depth analysis of (co)-regulation of transcripts and proteins across scales.

## Results

### A pipeline for deep spatial transcriptomic and proteomic profiling of the mouse hippocampus

We developed a pipeline for deep spatial transcriptomic and proteomic profiling of the mouse hippocampus from acutely isolated tissue, alleviating technical challenges inherent to approaches requiring tissue fixation. Performing manual dissections under a dissection microscope, we adapted a previous rat hippocampus dissection protocol for CA1, CA3, and DG[15] to the mouse hippocampus (as they are visually indistinguishable under a dissection microscope, CA2 and CA3 were dissected together). Using visual markers of the discrete layers, the four strata of CA1 (SO, SP, SR, and SLM) were then further dissected. For each sample, tissue from two mice was pooled to provide sufficient material for downstream analysis of the transcriptome and proteome from the same source, reducing technical and experimental variability. This approach enabled us to obtain sufficient material for parallel processing and analysis of both mRNA and proteins from the same tissue (Fig. 1a). We confirmed the accuracy of our dissections via detected enrichment of known molecular markers for each area; this included *Fibcd1* and Homer2 in CA1, *Bok* and Nectin3 in CA3, and *Prox1* and Calb2 in DG (Supplementary Fig. 1a)[7,48–56]. Markers for CA2 were additionally confirmed in CA3 samples (Supplementary Fig. 1b). Enriched markers in CA1 strata correspond to a combination of their known interneuron, glial, and synaptic populations: *Trhde* and Chrm2 in SO, *Cck* and Pvalb in SP, *Map2* and Lrrtm1 in SR, and *Ndnf* and Adgrl2

in SLM (Supplementary Fig. 2). These data demonstrate that this workflow identified spatial distribution of molecules from precisely microdissected hippocampi.

### Transcriptomic and proteomic landscapes of hippocampal subregions show overlapping patterns

Using our tissue microdissection and preparation pipeline (Fig. 1a), we profiled the transcriptomes and proteomes of subregions CA1, CA3, and DG. This workflow achieved deep coverage, quantifying >10,000 protein groups across subregions and detecting >17,000 mRNA transcripts (Fig. 1c,d). While all transcripts and the vast majority of proteins (~93%) were detected in all subregions, we asked if the local populations have different features using an unsupervised Principal Component Analysis (PCA), where the strong clustering of each subregion indicated that they are molecularly distinct (Fig. 1d). Along with the aforementioned enrichment of known subregion markers (Supplementary Fig. 1), differential expression of thousands of mRNAs and proteins was detected between subregions (Fig. 1e,g, Supplementary Datas 1-2), all of which can be found on our searchable platform https://syndive.org. We further validated key candidates for each subregion with in situ hybridization and immunofluorescent staining (Supplementary Fig. 3).

To explore the enriched molecular populations in each subregion, we performed Gene Ontology (GO) analysis to determine whether these transcripts and proteins overrepresented different functional roles (Fig. 1f,h). In CA1, both mRNA and protein profiles were enriched for postsynaptic and synaptic integration terms. This corresponded to enrichment of proteins such Shank3, Homer2, as well as GABA receptor subunit Gabra3, and glutamate receptor subunit Grin3a as well as the serotonin receptor subunit *Htr1b* and acetylcholine receptor subunit *Chrna5*. In CA3, GO terms highlighted axonal transport and myelin-related processes, along with enrichment of potassium channels such as KCNQ (including the protein encoded by *Kcnq2* and the *Kcnq5* mRNA) and KCN (including protein encoded by *Kcnc1* and the *Kcnc2* and *Kcna6* mRNAs) that regulate neuronal excitability[43]. For DG, the top terms were largely related to nuclear and gene regulation processes for both mRNA and protein, with mRNA processing as the top term in both datasets. Underlying molecules included mRNAs for transcription factors (*Prox1, Zpfm2*) and Wnt signaling (*Ctnnb1, Wnt16, Wnt9b, Axin2, Dvl2*) which supports neurogenesis in this region. At the protein level, the transcriptional regulator Mecp2 was another notable candidate. Mecp2 is mutated in Rett syndrome and in MECP2 duplication syndrome it disrupts DG neurogenesis[57,58].

### Strata-specific transcriptomes and proteomes reveal compartmentalized molecular signatures and functional roles

We next explored how these enrichment patterns might differ across the compartments of the CA1 pyramidal neuron. Given their strata-specific architecture, we examined how mRNAs and proteins contribute to the functional specialization of the pyramidal cell somata (SP), basal dendrites (SO), apical dendrites (SR), and distal tufts (SLM) and their associated interneuron populations (Fig. 2a). By analyzing over 17,000 mRNA transcripts and ~10,000 proteins (Fig. 2b), we identified distinct expression patterns that correspond to the specific functions of each CA1 stratum (Fig. 2), and validated key candidates (Supplementary Fig. 4). This stratified molecular organization, exemplified by clustering in PCA (Fig. 2c), revealed significant enrichment of thousands of mRNAs and proteins (Fig. 2d, f, Supplementary Data 1-2). GO analysis further indicated that the spatial regulation of these molecules gives rise to both formation of local microenvironments and specialization of neuronal compartments (Fig. 2e, g).

In SO, the detected enrichment of *Trhde*, *Erbb4*[59], and Chrm2[60] marks prominent interneuron populations. Other enriched mRNAs encoded proteins involved in trafficking and transport, such as kinesins and P4-ATPases, with associated GO terms linked to microtubules

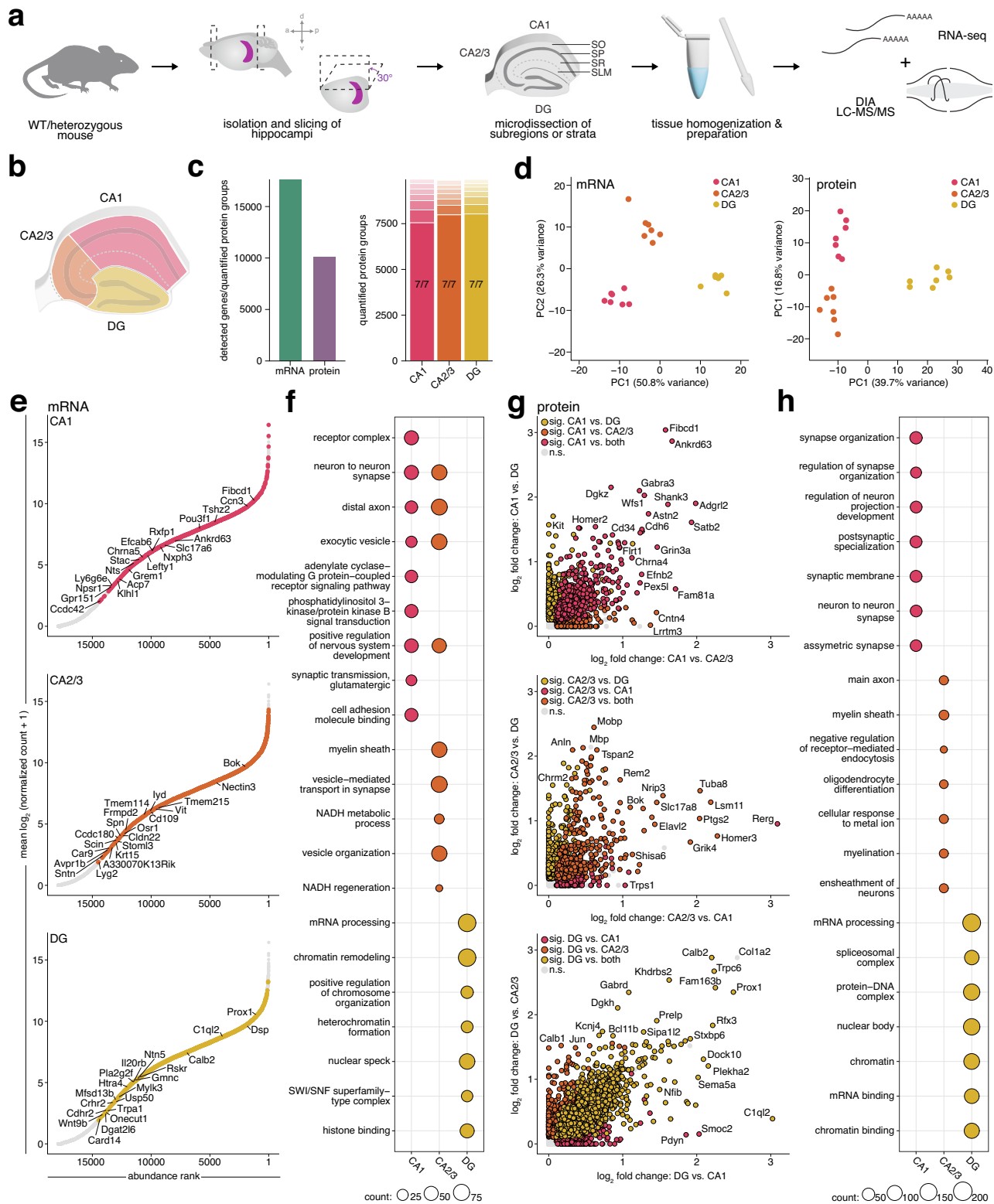

and axonal transport (Fig. 2e, Supplementary Data 1-2). We also found highly enriched myelin-related proteins (Mobp, Cnp, Cldn11, Mog, Ermn, Mbp, Pllp), which collectively contribute to the formation and maintenance of the myelin sheath[61,62]. These findings were consistent with significant GO terms related to axons, myelin, and gliogenesis (Fig. 2g, Supplementary Data 1-2). This aligns with the large population of myelinated pyramidal cell axons traversing the neighboring alveus, as well as oriens/alveus-bordering interneurons contributing to inhibitory circuits that regulate hippocampal network activity.

In SP, which contains pyramidal cell bodies, observed enrichment of *Cck* transcripts and Pvalb proteins marks basket cell populations that form perisomatic connections with pyramidal cells[60,63–67]. Despite being a dominantly somatic layer, GO term analysis revealed a strong presence of mRNAs related to neuronal and synaptic membrane organization, especially receptors and ion channels, indicating their synthesis in the cell body[19] before transport to their site of action (Fig. 2d, e, Supplementary Data 1-2). This included molecules linked to neurological disorders, such as *Grin2b* and *Grin2a* (epilepsy and

**Fig. 1 | A pipeline for deep spatial transcriptomic and proteomic profiling of the mouse hippocampus, and comparative analysis of subregion transcriptomes and proteomes. a** Experimental workflow for tissue analysis. Horizontal brain slices were generated after cutting hemispheres at a 30° angle and subregions or CA1 strata were microdissected and homogenized, followed by parallel RNA-seq and data independent acquisition (DIA) mass spectrometry (see methods). Tissue pooled from 2 mice represented 1 independent biological replicate. SO, *Stratum oriens*; SP, *Stratum pyramidale*; SR, *Stratum radiatum*; SLM, *Stratum lacunosum-moleculare*. **b** Schematic indicating subregions CA1, CA2/3 and DG that were microdissected for tissue preparation. **c** Left: total number of detected mRNA transcripts and quantified protein groups across all subregions (*n* = 7 replicates). Right: valid value bar plots of quantified protein groups per subregion. Opaque bars indicate proteins quantified in 7/7 replicates. **d** Principal Component Analysis (PCA) of transcriptome (left) and proteome (right) shows clustering by subregion. Each data point represents one replicate. **e** Rank abundance plots of mRNA indicating the relative abundance of transcripts in a given subregion. Transcripts with the highest log$_2$ fold change are ranked as 1. Coloured points represent transcripts significantly enriched in one subregion compared to the others (*s* value < 0.01, *n* = 7; see methods for details on normalization and differential expression), while all other detected transcripts are shown in gray. Top 10 transcripts by log$_2$ fold change in each comparison as well as known markers are labelled. **f** Gene ontology (GO) overrepresentation analysis based on significantly enriched transcripts from each subregion when compared to both others. Top terms were determined by a one-sided hypergeometric test (*p* adjusted <0.05). Bubble size corresponds to the number of genes annotated to a given term. **g** Scatter plots showing log$_2$ fold enrichment of proteins in each subregion compared to both others. Points are colored by significance (*p* adjusted <0.01, *n* = 7; see methods for details on normalization and differential expression) in each comparison. Non-significant proteins detected with 2 or more peptides are shown in grey. **h** GO overrepresentation analysis based on proteins significantly enriched in each subregion when compared to both others. Top terms and bubble size were determined as in (**f**). Source data are provided as a source data file.

intellectual disabilities[68]) and *Gabrb2, Gabrb3*, and *Cacna1e* (developmental disorders with epilepsy[69]). Since SP houses the pyramidal neuron nuclei, the site of transcription for all neuronal mRNAs (except those encoded by mitochondrial DNA), the abundance of synaptic mRNAs makes sense. Conversely, SP-enriched proteins, such as key transcription factors Neurod2 and Neurod6 involved in brain development and connectivity[70–73], were primarily linked to the nuclear processes occurring in the cell body (Fig. 2f, g). This layer exhibited the most pronounced functional differences between its transcriptome and proteome, likely owing to its housing of the nuclei which transcribe the huge majority of neuronal mRNAs.

SR, the largest fraction of neuropil in CA1, consists mainly of apical dendrites that receive Schaffer collateral inputs from CA3 axons. GO term analysis of the mRNA and protein profiles highlights processes related to synaptic function and modification (Fig. 2e, g, Supplementary Datas 1-2). Enriched mRNAs include kinases (*Stk38, Pdpk1, Prkcb, Braf, Mapk9, Ttbk2, Pdk3*) that influence phosphorylation and are associated with the GO term "peptidyl-serine phosphorylation," a type of post-translational modification that regulates synaptic function and plasticity[74,75]. For example, phosphorylation of PSD-95 at Ser295 enhances its stability and accumulation at synapses, promoting synaptic strength and AMPA receptor recruitment[74]. In presynaptic terminals, Ser129 phosphorylation of alpha-synuclein plays a critical role in enhancing its presynaptic targeting, by promoting its interaction with presynaptic proteins such as Vamp2 and synapsin[75]. The transcriptome also reveals an enrichment of UPS family members (*Usp9x, Usp40, Usp34, Usp32, Usp24, Usp48, Usp31, Mindy2*), emphasizing protein turnover and regulation, and its role at synapses and in synaptic plasticity[76–83]. Dysregulation of the UPS, as seen in Alzheimer's disease, disrupts SR synaptic function, whereas modulation of UPS activity can restore long-term potentiation[76]. Proteome analysis further highlights autophagy-related processes, with enrichment of proteins like Atg5 that regulate synaptic vesicle turnover[84] and autophagy to support plasticity[85]. Overall, the SR transcriptome and proteome converge on protein regulation mechanisms essential for synaptic function and plasticity.

SLM receives direct inputs from brain regions such as the entorhinal cortex, nucleus reuniens, and inferotemporal cortex[86–88]. Its transcriptomic profile reflects this, highlighting cell adhesion, extracellular matrix remodeling, growth factor signaling, and metabolic activity (Fig. 2e). Enrichment of the *Ndnf* transcript here serves as a marker for neurogliaform GABAergic interneurons with dense arborizations in this layer[89–91]. SLM also contains distinct astrocyte subtypes[92] driving enrichment of associated transcripts *Slc1a3, Fam107a, Agt* and *Gfap*. While the top enriched proteins in SLM included receptors (Grm2,4,6,8) and adhesion molecules (Ntng1, Cd44, Fbln5), GO analysis yielded mainly metabolic terms (Supplementary Data 1-2), indicating that a large proportion of enriched proteins are mitochondrial. As mitochondria are known to display morphological differences between neuronal compartments, this corroborates orthogonal findings that in SLM, mitochondria are longer and occupy more dendritic volume compared to SR and SO[93]. This may reflect an increased metabolic demand needed to process and integrate inputs[94], or to fuel local protein synthesis at sites far from the cell body[95].

## Spatial targeting of transcripts and proteins reveals functional modules linked to neuronal signalling, metabolism and disease

Our analyses revealed that molecular diversity across CA1 strata facilitates the emergence of different characteristics, with spatial compartmentalization shaping processes such as neuronal signaling and metabolism. We further investigated how these molecular profiles define functional modules by examining the fold-change distribution of all molecules assigned to five GO annotations with a focus on membrane proteins: K$^+$ channel complex, Na$^+$ channel complex, Ca$^{2+}$ channel complex, GPCR activity, and mitochondria (Fig. 3a). For all ion channel classes, the mRNAs showed the overall strongest trends towards enrichment in SP, again consistent with SP's high concentration of pyramidal neuron nuclei, the site of transcription. K$^+$ channel proteins were depleted in SP, and had positively skewed enrichment distributions in all other strata. For example, Kcnj3, 4, 6, and 9 were enriched in SLM, and Kcna1, 2, 4, and b2 were enriched in SO (Fig. 3b). Co-enrichment of mRNA and protein was only observed for Kcnk 1 and 2 (in SLM), consistent with their co-expression in astrocytes[92,96]. For ion channel complexes that are permeable to Na$^+$ or Ca$^{2+}$, expression was more polarized, with Na$^+$- permeable channel complexes overall enriched in SO and depleted in SLM whereas Ca$^{2+}$-permeable channel complexes conversely skewed towards depletion in SO and enrichment in SLM. Proteins exemplary of this organization include the AMPA-type glutamate receptor subunit Gria4, which is enriched in SO and depleted in SLM, and the NMDA-type glutamate receptor subunits Grin2a and Grin2b that are both depleted in SO (Fig. 3b). This is consistent with known gradients along CA1 dendrites of potassium, sodium, and calcium channels[43–45,97,98] as well as general molecular gradients throughout the hippocampus[7]. For instance, the distance-dependent increase in AMPA receptor number which is highest in SR and lowest in SLM[99], is consistent with our data (Fig. 3b). In line with receptor distribution also following gradients within the brain, we found GPCR-associated proteins gradually shifted from primarily depleted in SO to enriched in SLM, exemplified by distribution of metabotropic glutamate receptors (Fig. 3b). GPCR mRNAs exhibited a less uniform distribution, with separate subsets showing either a strong depletion or a strong enrichment in both SP and SLM. Mitochondrial mRNAs followed similar patterns with clear divisions into enriched and depleted groups in SP and SLM. In contrast, mitochondrial proteins were almost exclusively enriched in SLM. This reflects

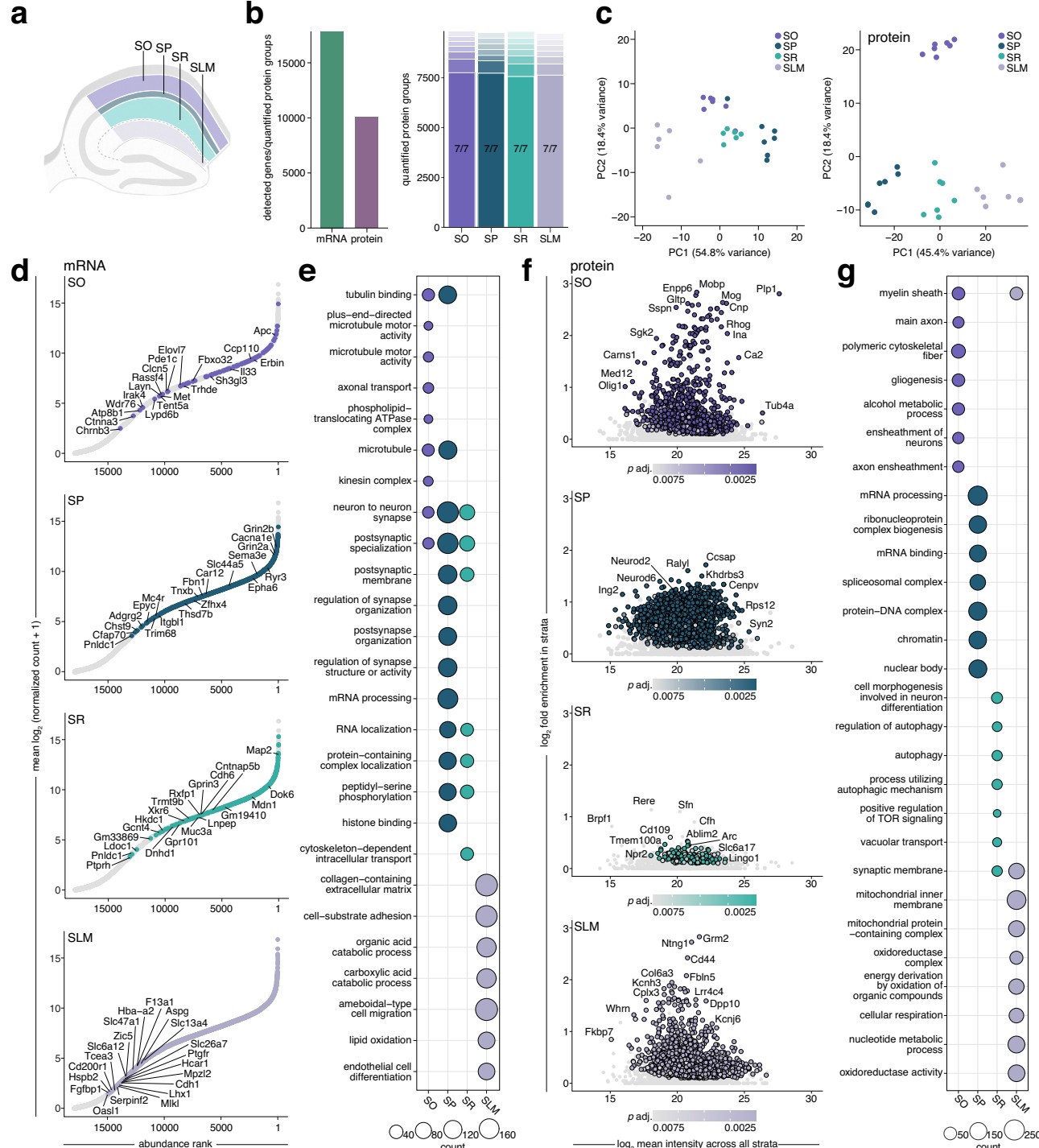

**Fig. 2 | Strata-specific organization of transcripts and proteins in CA1 give rise to specialized identities. a** Schematic indicating CA1 strata SO, SP, SR and SLM that were microdissected. **b** Left: total detected mRNA transcripts and quantified protein groups across all strata (*n* = 7 replicates). Right: valid value bar plots of quantified protein groups per stratum. Opaque bars indicate proteins quantified in 7/7 replicates. **c** PCA of transcriptomes (left) and proteomes (right) illustrating strata-specific clustering. Each data point represents one replicate. **d** Rank abundance plots of mRNA indicating the relative abundance of transcripts in given strata compared to whole CA1. Transcripts with the highest log₂ fold-change are ranked as 1. Coloured points represent transcripts showing significant enrichment (*s* value < 0.01, *n* = 7; see methods for details on normalization and differential expression), while all other detected transcripts are shown in gray. Transcripts with the top 20

highest relative enrichment (ordered by log₂ fold-change) are labeled. **e** Gene ontology (GO) overrepresentation analysis based on significantly enriched transcripts in each stratum. Top terms were determined by a one-sided hypergeometric test (*p* adjusted <0.05). Bubble size corresponds to the number of genes annotated to a given term. **f** Scatter plots showing log₂ fold enrichment of proteins in a given stratum compared to their mean log₂ intensity across all strata. Significantly enriched proteins (*p* adjusted <0.01, *n* = 7; see methods for details on normalization and differential expression) are shown in colour while all other enriched proteins detected with 2 or more peptides are in grey. **g** GO overrepresentation analysis showing top terms for proteins significantly enriched in each stratum. Top terms and bubble size were determined as in (**e**). Source data are provided as a source data file.

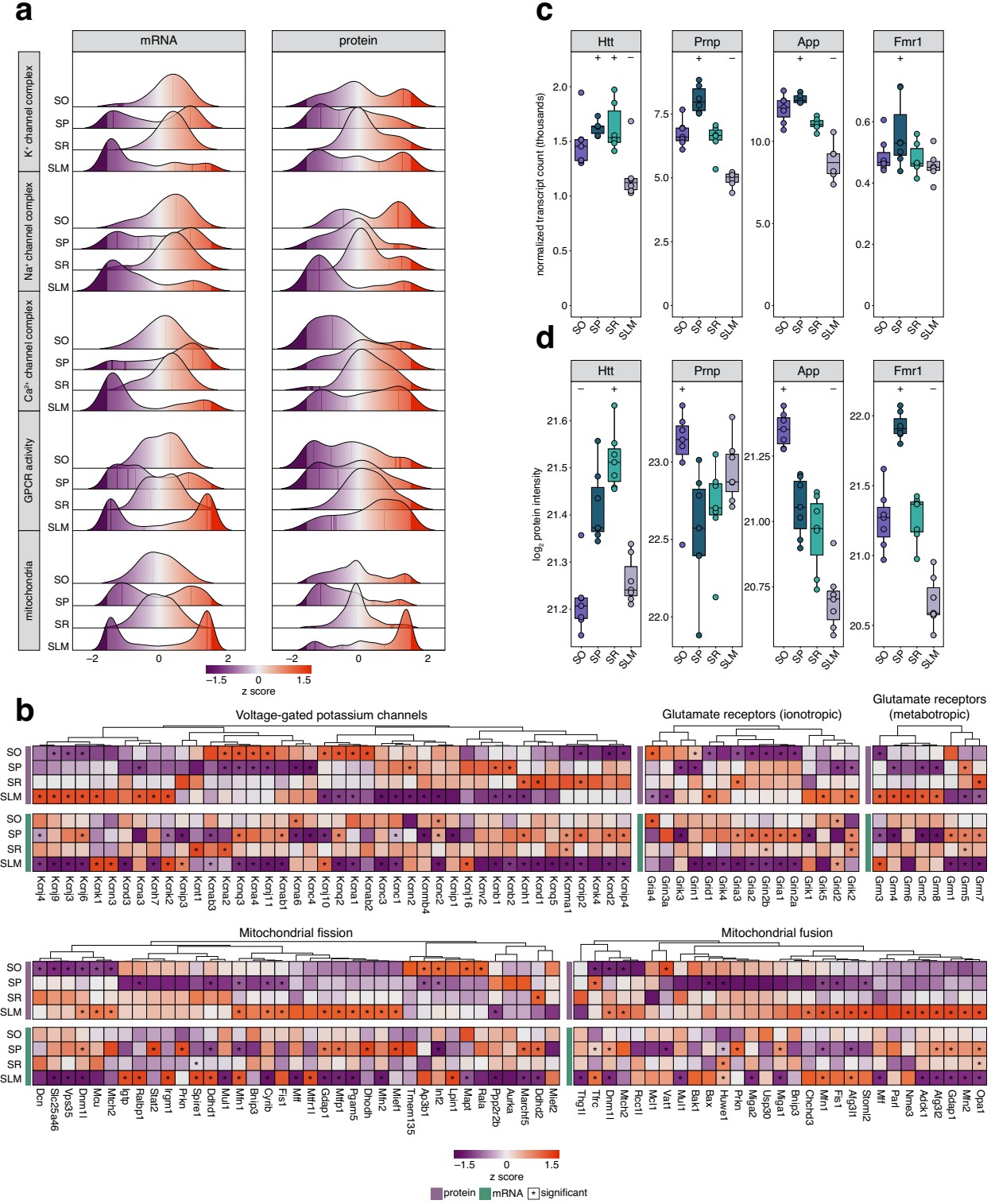

**Fig. 3 | Spatial organization of functional groups and disease-causing genes. a** Ridge plots of differential gene expression across CA1 strata for selected gene sets. Each plot represents the distribution of *z*-scored log₂ fold changes for all transcripts (left) or proteins (right) annotated to specific biological processes or cellular components (see methods, source data and Supplementary Data 1-2 for detailed annotations). **b** Candidate heatmaps for *z*-scored log₂ fold changes of transcripts and proteins associated with processes or cellular components shown in (**a**). Genes are ordered based on clustering of protein data. White boxes indicate missing values. Asterisks indicate significance in either mRNA or protein dataset as derived from differential expression analysis (s value < 0.01 or *p* adjusted <0.01, respectively, *n* = 7; see methods for details on normalization and differential expression). **c**, **d** Boxplots of abundances of mRNAs and proteins that have causative roles in disease and showed strata-specific enrichment. ± indicates significant enrichment or depletion in differential expression analysis (s value < 0.01 or *p* adjusted <0.01, respectively, *n* = 7; see methods for details on normalization and differential expression). Boxes indicate median (middle line) and interquartile ranges (IQR), with whiskers at 1.5 × IQR. Points represent values for each biological replicate (*n* = 7). Source data are provided as a source data file.

the high mitochondrial content of the distal dendrites[100] due to local fusion/fission dynamics that can be modulated by synaptic inputs and calcium signaling[101]. To further investigate this, we analyzed proteins involved in mitochondrial fusion and fission including the distribution of their respective mRNAs (Fig. 3b). Fusion regulators (Mfn1, Mfn2, Fis1, Mff) were indeed enriched in SLM, while regulators of both fusion and trafficking were SR-enriched (Dcn, Dnm1l, Slc25a26, Mcu, Ddhd1) and a mix of metabolic and quality control proteins were SO-enriched (Prkn, Inf2, Rala, Mapt, Lpin1, Inf2, Ap3b1, Tmem135, Igrp, Palbp1, Stat2, Irgm1, Prkn). Notably, Mfn1 protein and mRNA were co-enriched in SLM implicating a role for local translation.

The spatial organization of mRNAs and proteins across CA1 strata also has important implications for disease. Layer-specific vulnerabilities in the hippocampus have been associated with several neurological disorders[102], as molecular disruptions often disproportionately affect certain strata. For instance, Htt mRNA and protein, the key molecule of Huntington's disease, are enriched in the SR (Fig. 3c,d), where plasticity deficits and synaptic dysfunction are well-documented including SR-CA1 synapse impairments[103–105], and AMPAR diffusion dysregulation[106]. scRNA-sequencing studies have revealed significant transcriptomic changes in the hippocampus of Alzheimer's disease (AD) patients, particularly related to synaptic dysfunction, inflammatory responses, and changes to metabolism within SO and the adjacent alveus, where there were also disproportionately high levels of apoptosis[107]. This is consistent with previous findings that SO exhibits substantial inhibitory interneuron loss in AD, impairing feedback inhibition[108]. This disruption is compounded by well-documented damage to axons and myelin[109], along with the accumulation of misfolded prion proteins[110], and amyloid-β peptides, with SO disproportionately affected[111]. Our detected enrichment of Prnp and App in SO (Fig. 3c,d) aligns with these known disease phenotypes[107,108,110,111]. *Fmr1* mutations define Fragile X syndrome. In our dataset, Fmr1 protein and mRNA are most abundant in SP (Fig. 3c,d), in line with its role in translation as a translational repressor. Detection of *Fmr1* mRNA in other compartments in our dataset corresponds to its additional role as an RBP involved in mRNA trafficking[112]. Disruptions of this process as seen in *Fmr1* knockout cells impair synaptic mRNA targeting and local protein synthesis[113], and these disruptions may underlie the commonly observed spine abnormalities[114]. Overall, these examples highlight the importance of understanding the layer-specific distribution of molecules in CA1, as it may reveal underlying vulnerabilities for neurological disorders[102].

### Abundance comparisons of mRNA and protein unveils subcellular relationship patterns and dynamics

As we observed candidate mRNAs and proteins with both overlapping and opposing distribution patterns across area CA1, we further integrated the datasets to observe broader trends in the relationship between these molecules. We first asked whether individual mRNAs and their corresponding proteins are correlated in abundance, and whether this varied by strata. In line with previous reports[42], we found modest correlations in mRNA and protein population abundances across all strata ($r = 0.33–0.38$) with the highest correlation found in SP (Fig. 4a). To then deduce the extent to which variation in protein levels between strata might be due to variations in the corresponding mRNA, we correlated changes in relative mRNA-protein abundances across strata (Fig. 4b). This revealed subsets of hundreds of highly positively ($r \geq 0.9$, 1033 genes) and negatively ($r \leq -0.9$, 640 genes) correlated pairs (Supplementary Data 3). Positively correlated examples included the synaptic scaffolding protein Dlg1, showing the highest enrichment in SLM, and the RNA-binding protein Caprin1 and RNA-splicing protein Rbfox1, which both had the highest relative abundance in SP. Negatively correlated pairs included the proteasome subunit Psma1, and ribosomal proteins Rpl26 and Rps8 (Fig. 4b). To explore the biological

processes and molecular functions associated with all positively and negatively correlated pairs, we performed a GO overrepresentation analysis. Positively correlated mRNAs and proteins were enriched in protein-RNA regulatory complexes and processes related to neuronal structure and dynamics (Fig. 4c, Supplementary Data 3). This suggests that, for these groups, their relative protein abundances across strata are tightly linked to mRNA availability. By contrast, negatively correlated pairs were primarily ribosome associated, along with genes related to synaptic vesicle dynamics, protein degradation, and myelination (Fig. 4c, Supplementary Data 3). These molecules are likely regulated post-transcriptionally or post-translationally, with protein levels decoupled from mRNA. Protein turnover may also influence this relationship, as proteins with varying half-lives exhibit different mRNA-protein dependencies; for example, dendrite-associated proteins have shorter half-lives compared to those involved in ribosomal- or proteasomal- functions[115–118]. We explored this hypothesis by comparing positively and negatively correlated mRNAs and proteins to a dataset describing protein half-lives in hippocampal neurons[118], mapping a total of 2956 genes to their protein half-lives. Our analysis indicated that positively correlated mRNA-protein pairs have significantly shorter protein half-lives (Fig. 4d, Supplementary Data 3), suggesting that proteins with a more rapid turnover could be sustained by local translation of their cognate mRNA. In contrast, negatively correlated proteins had significantly longer half-lives, indicating that their levels are more stable and less influenced by local mRNA availability (Fig. 4d, Supplementary Data 3).

### Tracing mRNA localization and protein distribution across CA1 strata and potential hotspots for local translation

A central question emerging from these findings is to what extent mRNA levels and localization predict the compartment-specific distribution and abundance of their encoded proteins, and whether the co-enrichment of an mRNA and its respective protein is different across compartments. To investigate this, we systematically assessed the degree of overlap between enriched mRNAs and their corresponding proteins both within and across CA1 strata. This analysis assumed that co-enrichment of mRNA and protein in one compartment strongly suggests translation there, while mRNA enrichment in SP and protein enrichment in any other strata is consistent with synthesis in the cell body and subsequent trafficking. Our analysis revealed several key patterns (Fig. 5a, Supplementary Data 3). First, the greatest enrichment overlap was found in SLM, implicating the most distal dendrites as a hotspot for local translation. The second largest overlap was found for SP, where genes showing co-enrichment were largely related to processes such as mRNA processing and nuclear export. Third, SP-enriched mRNAs also mapped to proteins enriched in SLM, many of which were membrane proteins known to undergo extensive Golgi processing prior to transport, such as receptors and ion channels[119,120].

To test whether we could, in fact, attribute these enrichment patterns to local synthesis versus transport, we compared our dataset to previously published data on the translatome of CA1 somata (SP) versus neuropil (SR, SLM) from our laboratory[19]. Indeed, we found that 45% of mapped genes with co-enriched mRNAs and proteins in SLM are preferentially translated in the neuropil, whereas only 6% are predominantly translated in the cell body (Fig. 5b, Supplementary Data 3). Furthermore, 65% of SP-SP co-enriched mRNA-protein pairs and 71% of SP-SLM enriched mRNA-protein pairs are preferentially translated in the soma, suggesting that these mRNAs are translated in SP prior to trafficking of the protein. Together, these findings highlight a spatially regulated relationship between mRNA localization and protein synthesis in CA1, with distinct mRNA localization and translation patterns shaping the protein distribution across neuronal compartments.

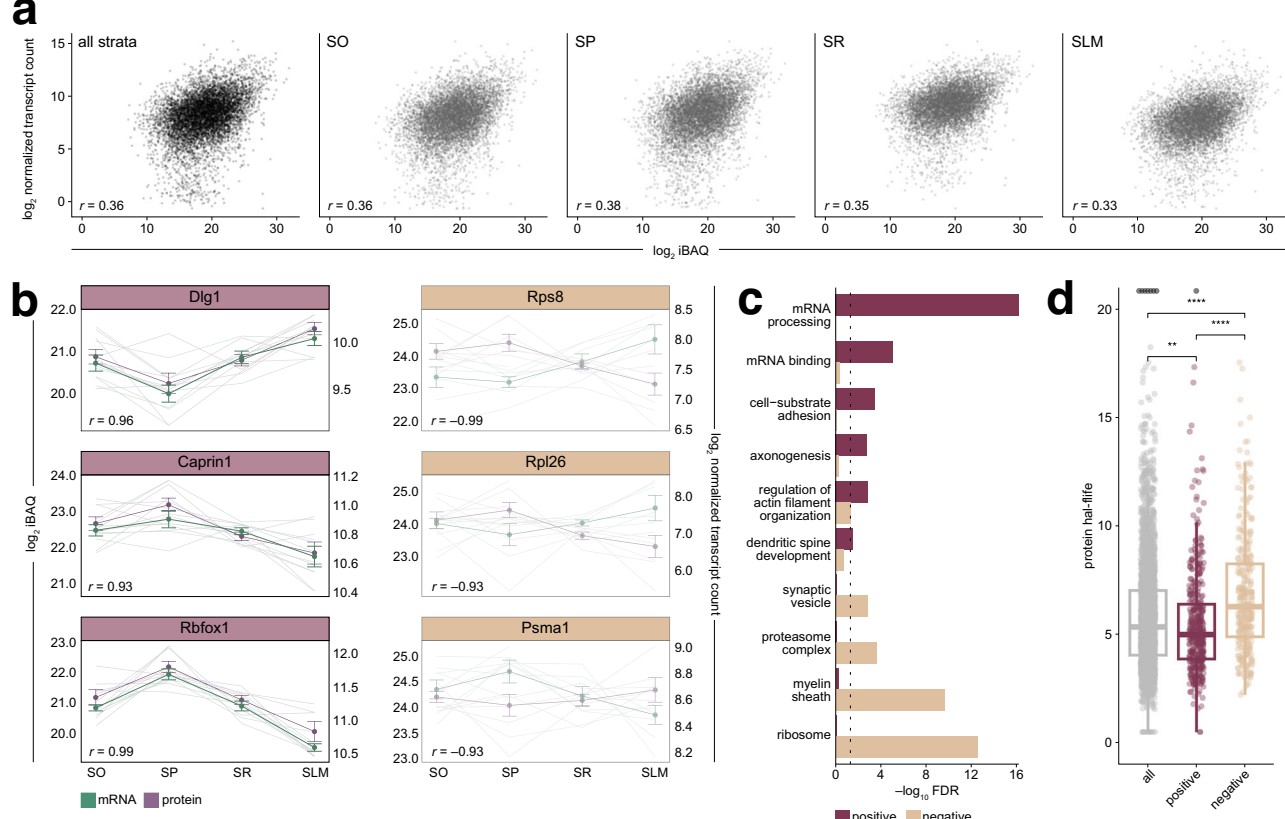

**Fig. 4 | Correlation analysis of mRNAs and proteins across pyramidal neuron compartments. a** Pearson's correlation of mean abundances of mRNAs (log$_2$ DESeq2 normalized counts) and their respective protein (log$_2$ intensity-based absolute quantification, iBAQ) across CA1 strata and within each stratum. **b** Selected examples showing strong positive (≥0.9) and negative (≤ −0.9) Pearson's correlation in their mean abundance changes across strata. Abundances for individual replicates are shown as translucent lines, while solid lines represent mean values with error bars indicating ± SD ($n = 7$). **c** Selected GO terms overrepresented by all transcripts and proteins showing a strong positive or negative correlation in their mean abundance changes across strata. Significance was determined by a one-sided hypergeometric test and -log$_{10}$ FDR values are indicated on the $y$-axis.

**d** Protein half-lives of strongly positively and negatively correlated mRNA-protein pairs. A total of 2946 genes were mapped to neuronal protein half-life data from Dörrbaum et al.[118]. Boxes indicate median (middle line) and interquartile ranges (IQR), with whiskers at 1.5 × IQR. Black translucent dots indicate outlier data points that fall beyond $y$-axis limits. Significance is indicated by asterisks (**$p < 0.01$, ****$p < 0.0001$) and was determined by Kruskal–Wallis test ($\chi^2(2) = 57.8$, $p = 2.84 \times 10^{-13}$, df = 2, $N = 3692$) followed by a post hoc Dunn's test with Bonferroni correction: all vs. positive ($Z = −3.12$, $p = 0.00181$, $p$ adjusted = 0.00542), all vs. negative ($Z = 6.54$, $p = 6.24 \times 10^{-11}$, $p$ adjusted = 1.87 × 10$^{-10}$), and positive vs. negative ($Z = 7.33$, $p = 2.29 \times 10^{-13}$, $p$ adjusted = 6.87 × 10$^{-13}$). Source data are provided as a source data file.

## Synaptic diversity across hippocampal subregions is shaped by different neuronal populations and connectivity profiles

To better understand the spatial organization of synaptic mRNAs and proteins, we applied a synapse purification strategy to hippocampal tissue and strata. Pan-synaptic Syn-1 labeling was used to capture synapses across subregions and strata, reflecting contributions from multiple cell types. To isolate synaptosomes from hippocampal tissue, we employed FASS[2,5,47] to isolate and purify synapses. Using this workflow in combination with our microdissection of acute tissue, we successfully purified synaptosomes from hippocampal subregions (CA1, CA2/3, DG; Fig. 6a) and strata (SO, SP, SR, SLM; Fig. 7). We sorted ~2 million synaptosomes from each microdissection to high purity, along with an equivalent number of control particles, despite limited starting material (Supplementary Fig. 5, see methods). After filtering for residual contaminant molecules enriched in a precursor fraction, the number of quantified proteins and detected mRNA transcripts across subregion and strata synapses totalled >5000 and 15,000, respectively, with high reproducibility between replicates (Figs. 6c and 7b, Supplementary Figs. 6-7, Supplementary Data 2 and 4).

To determine whether synapses within the hippocampus can be segregated based on their transcriptomes and proteomes of origin, we applied Partial Least-Squares Discriminant Analysis (PLS-DA). This classification method successfully separated CA1, CA2/3 and DG based

on both their synaptic mRNA and protein profiles (Fig. 6d). Top loadings across the first three components, presenting the most influential features driving this divergence, covered a range of molecules that inhabit both pre- and postsynaptic compartments (*Arhgap32*, Atg9a, Homer3, Shank1) in both excitatory and inhibitory synapses (Gria3, Gabra3, *Gabrr3*) (Supplementary Fig. 8). This indicates that transcriptomic and proteomic variability diversifies the synaptic populations within the hippocampus. Further analysis of the loadings revealed that specific molecular markers were particularly important in segregating certain subregions. For example, when the efficiency of loadings to differentiate one synaptic population from another was assessed using Variable Importance Projection (VIP) scores, we found that postsynaptic proteins were key in differentiating CA1 and CA2/3 from DG. Homer3, for example, had the overall highest VIP score for CA2/3, owing to its strong enrichment in this region, while Shank1 exhibited a high VIP score for DG, reflecting its relative depletion there (Supplementary Fig. 8). We subsequently compared the abundances of all transcripts and proteins across subregions to identify additional molecules with pronounced expression patterns. Region-specific specialization of inhibitory circuits[121] was illustrated by strong enrichment of *Gabrr3* in CA2/3 and Gabra1 in CA1. Molecular signatures defining hippocampal pathways were also differentially enriched. Among these are Leucine-rich repeat (LRR)-containing synaptic adhesion molecules,

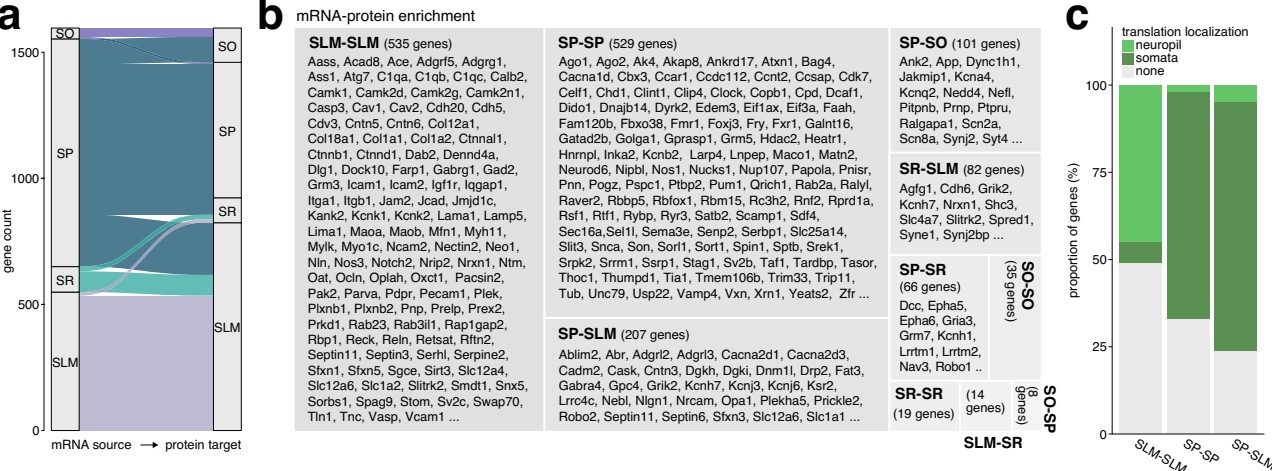

**Fig. 5 | Comparing mRNA localization and protein distribution across CA1 strata. a** Alluvial plot showing the site of enrichment of mRNAs and their corresponding proteins for different combinations of SO, SP, SR, and SLM. Width of flows represents the number of overlapping genes. **b** Tile plot showing candidate genes based on their mRNA-protein enrichment (also see Supplementary Data 3). Gene number represents the number of overlapping enriched mRNAs and proteins for a given combination. Tile size is proportional to overlap count. **c** Proportions of mRNAs with preferential translation in somata or neuropil based on the enrichment site of protein they encode. Genes were mapped to translatome data from Glock, Biever, et al.[19]. Translation localization was determined on the basis of genes showing significant enrichment in somatic versus neuropil compartments in the translatome dataset[19]. Source data are provided as a source data file.

which play a crucial role in organizing synaptic specificity and diversity (for reviews see refs. 122,123). For example, we found the enrichment of mRNAs of *Lrrtm4* and its interactor *Gpc4*[124] which have been implicated in synaptic patterning from the lateral entorhinal cortex to distal dendrites of DG granule cells[122,125]. We also observed enrichment of *Lrrtm1* in CA1, consistent with its postsynaptic localization at Schaffer collateral synapses[126]. Together these findings highlight the distinct molecular profiles of hippocampal subregions, emphasizing circuit-specific mechanisms and the critical role of adhesion molecules in circuit organization.

## Classification of compartment-specific synaptic mRNA and protein signatures along the pyramidal neuron axis

Synapses have traditionally been classified based on neurotransmitter type, but it is now recognized that there is greater diversity in molecular composition[5,127–129]. We hypothesized that the spatial distribution of synapses may further contribute to this complexity. CA1 pyramidal neurons are composed of different presynaptic inputs that originate from both within and outside the hippocampus. By combining our microdissections of CA1 with FASS (Fig. 7a, b), we asked whether these synapses differ despite converging on one neuronal population. Employing PLS-DA, we found that the transcriptomes of these synapses were sufficiently diverse to separate them by strata (Fig. 7c). Previous work from our group has shown that the proteomes of one synapse type are highly similar within one brain region[5]. Classification of the proteomes of pyramidal neuron synapses recapitulated this, with PLS-DA most clearly distinguishing SLM synapses, which receive thalamocortical inputs, from all other strata, where most inputs originate from within the hippocampus. Modest separation of SR synapses, receiving inputs from CA3, was still achieved, whereas SO and SP synapses intermingled.

By extracting the top loadings from PLS-DA, we then asked what types of molecules are most critical in defining these spatially restricted subpopulations (Fig. 7d). Many of the identified mRNAs were associated with kinase activity (*Camk2b, Grk2*), and synaptic remodeling during neurotransmission (*Cnih2, Gabbr3, Dock10*). In contrast, key proteins were involved in cell adhesion (Afdn, Cadm2, Plxna4) and cytoskeletal organization and transport (Actn1, Coro1a[130], Coro2b[131], Kif5c). The VIP scores for these molecules, which indicate how well they differentiate one synaptic population from another, also differ by

strata (Fig. 7d). For instance, *Cnih2* mRNA or the Actn1 protein show high VIP scores in SR compared to other layers, suggesting that they may play key roles in synaptic remodeling in this stratum.

To further explore these molecular classes, we examined key transcripts encoding kinases and other synaptic signaling molecules (Fig. 7e). Spatial enrichment patterns were found for Ca²⁺/calmodulin-dependent kinase, mitogen-activated protein kinase, and serine/threonine protein kinase transcripts, along with mRNAs encoding regulators of synaptic vesicle exocytosis (RIMs) and scaffolding proteins (DOCK, DLGAP, DLG families) (Fig. 7e). Previous studies of glutamatergic synaptosomes showed enrichment of mRNAs related to neurotransmitter release, such as RIMs, which we observe a similar trend for specifically in SLM (Fig. 7e). Kinases are also known to be locally translated at synapses where they directly modulate processes such as receptor clustering and ion channel expression[132], and their differential abundances consequently impact plasticity properties. We found that *Camk2n1, Mapk8ip1,* and *Prkacb* mRNA all preferentially localize to SLM synapses, whereas *Camk2a* and *Prkcb* mRNA are enriched at SR synapses. This corroborates previous evidence of their translation in these regions[19], and suggests that they may underlie the unique forms of plasticity that occur in these layers.

For the synaptic proteome, several members of adhesion, axonal guidance, and neuronal recognition families emerged as defining features. This encompassed cadherins, neuroligins, neurexins, ephrins and plexins, all of which are involved in synapse formation and remodeling[133–139] (Fig. 7e). Similar to kinases, these proteins confer plasticity properties. For example, the endocytosis of N-cadherin (Cdh2, here enriched in SLM) is crucial in linking synaptic activity to long-term changes in synaptic strength and structure[140]. Cdh6 and Cdh9 interact to mediate high-magnitude long-term potentiation and influence mushroom spine density, which is uniquely observed in SO; we observed a trend for Cdh6 and 9 enrichment in SO (Fig. 7e)[141]. Ephrins, such as Epha4, were specifically enriched in SR, and to a lesser extent in SLM, where they are likely to play a role in the organization and formation of dendritic spines[142]. Several molecules with enrichment trends were also mapped to innervating connections, such as Epha5 in SLM, which is involved in establishing the entorhino-hippocampal connection[143]. Presynaptic adhesion molecules Nrxn1 and Nrxn2 were also SLM-enriched, consistent with their expression in thalamic neurons that project to this layer[144]. Interestingly, we also

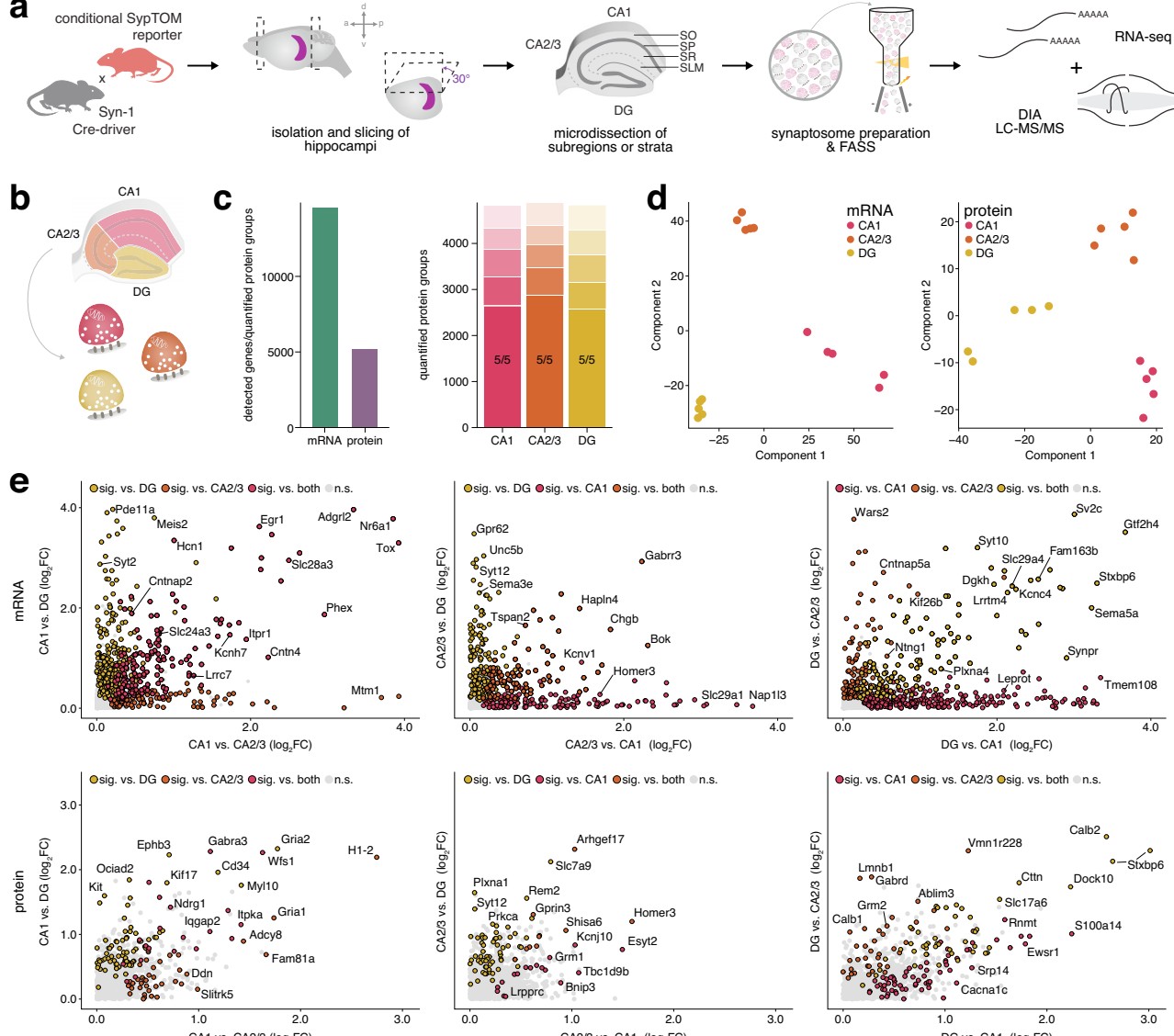

**Fig. 6 | Subregion-specific neuronal populations shape synaptic diversity in the hippocampus. a** Synaptosome isolation workflow: A *Syn-1 Cre*-driver line was crossed with a floxed Synaptophysin-tdTomato (SypTOM) reporter to label presynaptic terminals. Horizontal brain slicing and dissection was performed as in Fig. 1a. Crude synaptosomes were isolated using a Percoll-sucrose gradient and sorted using fluorescence-activated synaptosome sorting (FASS), followed by parallel RNA-seq and DIA mass spectrometry. Tissue pooled from 2 mice represented 1 independent biological replicate. **b** Schematic of synaptosomes from microdissected subregions. **c** Left: Detected transcripts and quantified protein groups in purified synaptosomes following contaminant filtering for each subregion (*n* = 5). Right: valid value bar plots of quantified protein groups per subregion. Opaque bars indicate proteins quantified in 5/5 replicates. **d** PLS-DA plot showing segregation by subregion for both transcriptome and proteome. Each data point represents one replicate. **e** Scatter plots of log₂ fold changes for transcripts (top) and proteins (bottom) showing positive enrichment in synaptosomes in each subregion versus the other groups. Points are colored by significance in each comparison (*s* value < 0.05 or *p* adjusted <0.05, respectively, *n* = 5; see methods for details on normalization and differential expression). Non significant points are shown in grey. Top candidates are labelled with their gene name. Source data are provided as a source data file.

---

observed differential expression of actin proteins across our dataset, with F-actin proteins Actn1 and Actn2 enriched in SR (Fig. 7e). It was previously shown that there is bidirectional regulation of actin proteins at dendritic spines, with a shift towards increased F-actin presence following tetanic stimulation[145]. Differential enrichment of kinesins and tubulins in our purified synaptosomes also suggests the potential for localized control over transport to and from specific synapses. The enrichment of Kif21a in SLM corresponds to a need for transport and insertion of GABA receptors[146] in this layer, whereas Kif5a enrichment in SR is likely linked to its role in transport of AMPA receptors[147]. Taken together, these results reveal a clear spatial component to synaptic diversity, characterized by distinct mRNA and

protein signatures for pre-, post- and trans-synaptic molecules along the CA1 pyramidal neuron.

## Discussion

Understanding the molecular diversity of the brain at multiple layers requires detailed maps of local transcriptomes and proteomes with spatial resolution and high coverage. Efforts to address this challenge have been hindered by the necessity to correlate transcripts from one sample or from one published study with proteins from another. Here, we microdissected hippocampal subregions and strata and performed FASS-based synaptosome purification for RNA-seq and LC-MS/MS analysis of the same samples in parallel. We

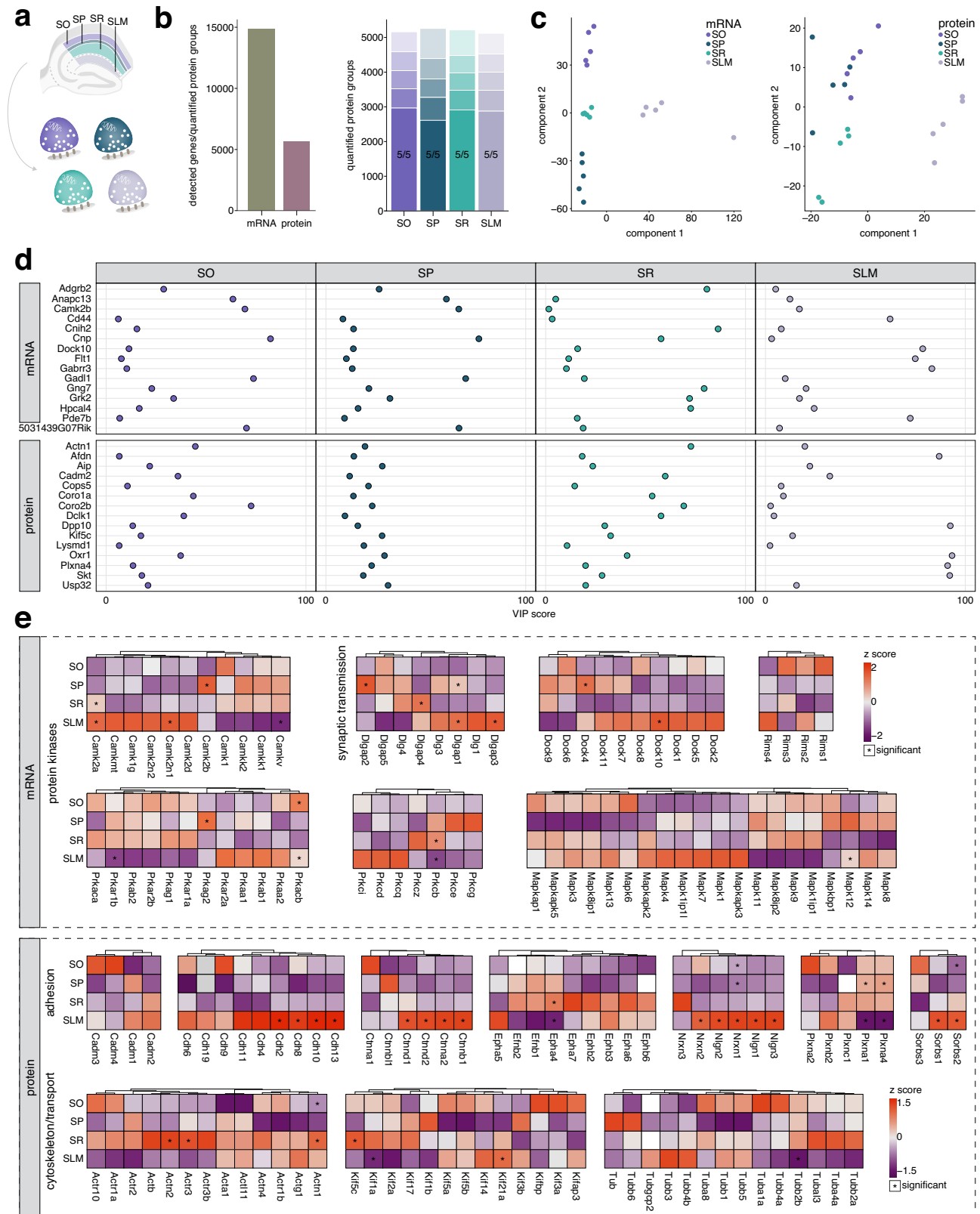

found marked differences in the transcriptomic-proteomic landscape of the different hippocampal subregions and CA1 strata defined by their composition of differing cell types, projections, and subcellular compartments. Synapses are traditionally classified by neurotransmitter type, but here we show deeper diversity in mRNA and protein composition along the pyramidal neuron revealing a spatial component to this diversity.

Our analysis of mRNA and protein from the same tissue minimized batch effects and cross-sample correlation discrepancies, enhancing reproducibility and reducing biological variability. Our precision dissection approach eliminated the need for fixation or laser-capture microdissection, preserving molecular integrity while maintaining spatial resolution. Unlike lower-coverage in situ techniques such as MALDI imaging of proteins or in situ sequencing, our method

**Fig. 7 | Classification of compartment-specific synaptic mRNA and protein signatures along pyramidal neurons. a** Schematic of synaptosomes from microdissected strata. **b** Left: Detected transcripts and quantified protein groups in purified synaptosomes following contaminant filtering for each stratum ($n = 5$). Right: valid value bar plots of quantified protein groups per stratum. Opaque bars indicate proteins quantified in 5/5 replicates. **c** PLS-DA plot showing segregation of strata for both transcriptome and proteome. Each data point represents one replicate. **d** Variable Importance in Projection (VIP) plots showing VIP scores for the top five loadings from the first three components in PLS-DA. VIP scores indicate the

contribution of each variable to the discrimination between strata in the PLS-DA model. Scores are scaled from 0 to 100. Higher VIP scores represent greater importance in distinguishing between strata. **e** Candidate heatmaps displaying the expression patterns of mRNA and proteins across synapses along the CA1 neuron. White boxes indicate missing values. Asterisks indicate significance in either mRNA or protein dataset as derived from differential expression analysis ($s$ value < 0.05 or $p$ adjusted <0.05, respectively, $n = 5$; see methods for details on normalization and differential expression). Source data are provided as a source data file.

improves molecular depth. The RNA-seq used here provided comprehensive transcriptomic profiles and single-shot MS runs identified ~10,000 proteins from tissue per subregion or strata, achieving coverage comparable to or greater than other studies performing fractionation-based proteomic profiling of the mouse and human hippocampus[3,148]. We successfully isolated high-purity synaptosomes from both hippocampal regions (CA1, CA2/3, DG) and precisely microdissected layers (SO, SP, SR, SLM), overcoming technical limitations of past approaches using rats[15,19] which widely lack available transgenic models. Using transgenic mouse lines allows for the targeted labeling of synapses and subsequent FASS, facilitating a thorough understanding of the molecular landscape underlying synaptic function and diversity. This high-resolution approach revealed differential expression of thousands of molecules, discovering previously undescribed spatial distribution of mRNAs and proteins across tissue and synaptosomes.

Our analyses revealed unique molecular signatures that align with hippocampal specialization. By comparing locally enriched mRNA and protein pools across the dendrites of CA1 pyramidal neurons, we found that ubiquitin-proteasome system transcripts and autophagy proteins, related to plasticity and synapse remodeling, are enriched in SR, whereas nearly all mitochondrial proteins (and a subset of mitochondrial transcripts) are enriched in SLM. These findings identify the layer-specific molecular regulation that underlies differences in hippocampal function and how its dysregulation may be involved in neurological disorders. Consistent with previous studies[14], we found a modest direct correlation in abundances of mRNAs and their protein products in all of the compartments we examined. Remarkably, there were still hundreds of transcript-cognate protein pairs with high positive or negative correlation in their abundance variation. By comparing these groups to protein turnover data, we found that positively correlated pairs had significantly higher protein turnover rates (shorter half-lives), whereas negatively correlated pairs had lower turnover rates (longer half-lives), suggesting that protein stability influences the degree to which the local protein pool will rely on local translation versus transport. To further investigate a role for local translation in the spatial targeting of proteins, we performed co-enrichment analysis of mRNAs and proteins across compartments. The largest mRNA-protein enrichment overlaps, ranging from 200–500+ genes, were found for SLM-SLM, SP-SP, and SP-SLM. By comparing with a hippocampal translatome dataset[19], we found that in ~50% of the cases where the mRNA and respective protein are co-enriched in SLM, the protein is preferentially translated locally. Conversely, mRNA enrichment in SP and the cognate protein enrichment in SLM was predictive of translation in the soma prior to transport. These results highlight the compartmentalized nature of molecular processes[17–20,42,149] and gradients across the hippocampus[7,48], supporting layer-specific molecular regulation, local translation, and protein trafficking, all of which contribute to hippocampal function.

Increasing evidence has uncovered considerable diversity in the structure, molecular composition and function of synapses[5,127–129]. Here, we show that the spatial distribution of synapses contributes further to this complexity and diversity, both across subregions (CA1, CA2/3, DG) as well as across the distinct synaptic inputs that converge

on a single region (SO, SP, SR, SLM). By examining compartment-specific synaptic transcriptomic and proteomic signatures from microdissected regions and layers of CA1 pyramidal neurons, we identified distinct, region- and strata-specific synaptic profiles. At the synaptic transcriptome level, we identified mRNAs coding for kinases and other proteins known to modulate synaptic function. These include components of signaling pathways, scaffolding molecules, and elements of the translational machinery itself. Our findings are consistent with earlier studies of glutamatergic synapses in the cortex[47], and they further support the growing recognition that both harbor a rich repertoire of distinct pools of localized mRNAs. This synaptic transcriptome enables rapid, compartment-specific protein synthesis, as demonstrated by Hafner et al.[47], who showed that local translation occurs ubiquitously in neuronal processes and plays a critical role in modulating synaptic plasticity and function. Many of these mRNAs were specifically enriched in SLM synapses, suggesting that synapses in this region differ significantly from those in other CA1 layers. This is supported by the fact that SLM receives distinct inputs from the entorhinal cortex, nucleus reuniens, and inferotemporal cortex[86–88], while SO, SP and SR predominantly receive inputs from within the hippocampus[150]. Interestingly, we found some mRNAs enriched in synapses that encode neuronal surface proteins such as ionotropic glutamate and GABA receptors, which are typically thought to undergo post-translational processing before reaching synapses[119]. However, our observations align with previous datasets showing that while these ionotropic and metabotropic receptor subunits are predominantly translated in the cell body layer, a small fraction is translated in the neuropil[19]. There, these mRNAs may be translated on-demand[22], potentially bypassing the Golgi apparatus to directly tune synaptic strength[120,151]. At the protein level, we similarly found that SLM synapses can be defined in part by their inputs, particularly via enrichment of presynaptic adhesion molecules that may be specific to entorhinal connections, such as Nrxn1 and Nrxn2[144]. Another adhesion molecule, Epha4, emerged as a defining protein of SR synapses. Plasticity properties of the layers could also be partially explained by the complement of enriched proteins, with F-actin molecules also enriched at SR where they support responses to high-frequency inputs that drive LTP[145].

Our study provides a comprehensive molecular map of hippocampal synaptic diversity, but there are several areas for future exploration. First, while we used Syn-1 to capture pan-synaptic data, future work could focus on distinguishing specific synapse types (e.g. excitatory vs. inhibitory) to capture diverse neuronal populations[152,153] within the hippocampus. Additionally, our analysis captures static molecular profiles at a single time point, and future research could explore how these profiles are dynamically modulated by activity or disease. Further investigation into regulatory mechanisms such as 5' and 3' UTRs, miRNAs, specific RNA binding proteins, RNA granules or other post-translational modifications would also add to a more complete picture. Our focus on hippocampal subregions may overlook broader network interactions with other brain regions. Lastly, while integrating multi-omics datasets remains challenging, our dataset provides a strong foundation for future efforts to build comprehensive models and, as such, this work lays the groundwork for future research on dynamic, network-wide and detailed molecular landscapes.

## Methods

### Experimental model and animal details

The procedures involving animal treatment and care were conducted in conformity with the institutional guidelines that are in compliance with national and international laws and policies (DIRECTIVE 2010/63/EU; German animal welfare law; FELASA guidelines). The animals were euthanized according to annex 2 of § 2 Abs. 2 Tierschutz-Versuchstier-Verordnung and animal numbers were reported to the local authorities (Regierungspräsidium Darmstadt). Mice were housed under a 12 h light/12 h dark cycle at 20–24 °C ambient temperature and 55 ± 10% relative humidity.

### Animals used for transcriptomics and proteomics

10/11-week-old *Cre*-positive and *Cre*-negative F1 littermates from offspring of a cross between hemizygous *Syn1-Cre* female mice (B6.Cg-Tg(Syn1-cre)671Jxm/J, RRID:IMSR_JAX:003966) and homozygous *condSypTom* males (B6;129S-Gt(ROSA)26Sortm34.1(CAG-Syp/tdTomato)Hze/J obtained from The Jackson Laboratory, RRID:IMSR_JAX:012570)[154] were used (deposited by Hongkui Zeng Lab). *Cre*-positive mice were used for the synaptosome generation and FASS and their *Cre*-negative aged-matched litter mates were used for whole tissue preparations and protein validation experiments. To allow enough material for downstream transcriptomic and proteomic analysis, hippocampi of two mice, one male one female, were pooled together for one replicate (*n* = 1, hippocampi of two mice) which was then divided following slicing and microdissection for parallel transcriptomic and proteomic analysis. Data are not separated by sex due to pooling. Inclusion of both sexes per replicate reflects combined variability, and prior analyses showed negligible sex-related variance[5].

### Tissue preparation and microdissection

Preparation of acute hippocampal slices from *Cre*-positive mice and their *Cre*-negative siblings was done as previously described[82]. Briefly, mice were decapitated after isoflurane anesthesia, the brain was immediately removed and put in ice-cold, slushed artificial cerebrospinal fluid (ACSF) solution with sucrose (87 mM NaCl, 25 mM NaHCO₃, 1.25 mM NaH₂PO₄, 2.5 mM KCl, 10 mM Glucose, 75 mM Saccharose, 7 mM MgCl and 0.5 mM CaCl₂) carbogenated with 95% O₂/5% CO₂. The brain was cut into halves along the longitudinal fissure, the neocortical side of the hemispheres were cut at an angle of 30° and glued on an ice-cold specimen holder and sliced with a Vibratome (VT 1200 S, Leica) to 300 μm-thick horizontal/transverse slices in ice-cold, carbogenated ACSF solution with sucrose. Slices from the middle part of the hippocampus were selected for dissection by clear visual identification of subregions and CA1 layers. Before slice preparation, blades were adjusted parallel to the cutting surface using VIBRO CHECK and set to a cutting speed of 0.08 mm/s with an amplitude of 1 mm. Generally, the left and right hippocampi taken together from one mouse yielded 8–12 optimal slices that were used for microdissection. Microdissections of CA1, CA2/3, DG and the *Strata oriens, pyramidale, radiatum* and *lacunosum-moleculare* were carried out manually under a dissecting microscope at 4 °C using a microdissection scalpel (McKesson,701459/Surgical Specialties, 7503; Stab Knife MSP™ Stainless Steel, Blade / Tip Type 15° Angled Straight Sharp Pointed Tip) and sections were collected in 0.5 mL RNAse-free 1 X PBS supplemented with Millipore Protease inhibitor cocktail III (539134) 1:750 and SUPERase•In™ RNase Inhibitor 1:80 (AM2696) in 1.5 mL LoBind Eppendorf tubes. In brief, the division of the hippocampal fissure serves as a visual separation between Ammon's horn (*Cornu Ammonis*) and the DG, facilitating the isolation of the upper segment of Ammon's horn (CA1). Removal of the subiculum from the whole brain slice facilitated access to CA1. Following this, the remaining portion of Ammon's horn (CA3 and CA2 together as they were visually indistinguishable) was separated from the DG, the boundaries of which are

optically identifiable, the hillus was cut from approximately the suprapyramidal blade to the infrapyramidal blade. After dissection, the combined sections from a given region-of-interest were immediately spun down, the supernatant was removed and the samples were snap-frozen and stored at −79 °C. This dissection technique provided sufficient material for both tissue and synapse purification without laser-associated heat degradation, tissue fixation or freezing[155], yielding higher amounts for downstream synaptosomes and tissue preparation.

### Percoll density gradient for crude synaptosome preparation

The same procedure, described above, for preparing hippocampal slices and performing microdissection was followed. Synaptosomes were prepared as previously described[5,156]. After collection in LoBind Eppendorf tubes, the samples were transferred to a clean 1.0 mL glass WHEATON® Dounce Tissue Grinder and homogenized using 20 strokes with the 'loose' and 20 strokes with the 'tight' pestle in gradient medium (GM; 0.25 M sucrose, 5 mM Tris-HCl, 0.1 mM EDTA supplemented with Millipore Protease inhibitor cocktail III 1:750 (539134) and SUPERase•In™ RNase Inhibitor 1:80 (AM2696)). The homogenate was centrifuged for ten minutes at 4 °C at 1000 × g. The supernatant (S1) was layered onto a Percoll density gradient with 23%, 10% and 3% Percoll in the GM buffer. The gradient was centrifuged for 5 min at 32,500 × g at 4 °C with maximum acceleration and minimum deceleration using a Beckman Coulter JA-25.50 rotor in an Avanti J-26S XPI centrifuge (both from Beckman Coulter). The resulting bands were labeled, from top to bottom, F0, F1, F2/3, F4. The F2/3 band (at the interface of 23% and 10%) was retrieved with a syringe and a blunt cannula and used as an input for subsequent FASS. All procedures were conducted on ice.

### Fluorescence-activated synaptosome sorting (FASS)

Dissections were performed and synaptosomes from each hippocampal region and CA1 strata from the *Cre*-positive mouse strain (see above) were isolated as described above[156]. Subsequent FASS was then used to sort synaptosomes to high purity[5] (Supplementary Fig. 2a, b) on a FACSAria Fusion (BD Biosciences) running FACSDiva (BD Biosciences). Doublet particles were excluded based on SSC-H and SSC-W and other parameters were as described[5] including the following settings: 488 nm laser (for FM4-64), 561 nm laser (for tdTomato signal), sort precision (0-16-0), FSC (317 V), SSC (488/10 nm, 370 V), PE "tdTomato" (586/15 nm, 470 V), PerCP "FM4-64" (695/40 nm, 470), thresholds (FSC = 200, FM4-64 = 700). Samples were analyzed and sorted at ~20,000 events/s and a flow rate of <5. BD FACSFlow (12756528) was used as a sorting buffer. Synaptosomes were sorted through a 70 μm nozzle and collected on a Whatman GF/F glass microfiber filter (1825-090) supported by a PE drain disk (231100) for stability. Both were punched out together using a Militex biopsy Punch (4 mm, 15110-40-X) which was washed prior to sorting with either 700 μL RNAsecure (1:5; Thermo Fisher Scientific, AM7006) or Millipore Protease inhibitor cocktail III (1:750; 539134) to inhibit RNAses or Proteases in the sorting buffer, respectively. After 2 million synaptosomes were collected on the filter, it was washed with 1 mL RNAse-free 1 X PBS to remove residual FACSFlow and stored at −79 °C until further processing. Purified synaptosome (here referred to as P3 based on the gates used in the sorting layout) fractions were sorted based on the presence of both a membrane dye (FM4-64; Thermo Fisher Scientific, T13320) and the tdTomato signal. Control (here referred to as P2 or synaptosome-sized) fractions were also generated from the F2/3 precursor synaptosome fraction based on only the membrane dye signal intensity but independent of tdTomato signal, resulting in a conservative unsorted fraction which contained both synaptosomes and synaptosome-sized contaminants. Synaptic enrichment was later determined by comparing abundances in P3 relative to P2 fractions. After FASS,

filters with synaptosomes were stored at −79 °C until further processing.

## Tissue homogenization, RNA extraction, library preparation and sequencing

Tubes with tissue samples were kept on dry ice until RNA extraction. Frozen tissue samples were resuspended in 1 mL TRIzol, transferred to a clean 1.0 mL glass WHEATON® Dounce Tissue Grinder and homogenized using a 'loose' and 'tight' pestle for 20 strokes each. The homogenate was then transferred back to the 2 mL RNAse-free Eppendorf tube and triturated 20x using a 23G needle and 1 mL syringe[19] and kept on ice. All samples were incubated for 5 min at room temperature and centrifuged for 3 min at 16,100 × g. The clean supernatant was collected in a new RNase-free tube and RNA was column purified using ZymoReaserach kit. The RNA from CA1, CA2/3 and DG subregions was extracted using the Direct-Zol RNA Miniprep Kit (R2051) and from the CA1 *Strata* using the Direct-Zol RNA Microprep Kit (R2061). RNA quantity and quality were measured on a Qubit fluorometer (Invitrogen, Q33216) and on a 2100 Bioanalyzer (RNA Pico chip, Agilent Technologies).

mRNA-seq libraries were prepared from 25 ng of total RNA using NEBNext Ultra II Directional RNA Library Prep Kit for Illumina (E7760), Poly(A) mRNA Magnetic Isolation Module (E7490) and Multiplex Oligos for Illumina Set 1 and Plate IDs: C7-H7, A8-H8, A9-H9, A10-H10, A11-H11, A12-H12 (E6440) (see Supplementary Data 5). The quantity of the libraries was measured on Qubit fluorometer (Invitrogen, Q33216) and quality assessed using HSDNA 2100 Bioanalyzer assay (Agilent Technologies). Libraries were sequenced on Illumina NextSeq2000 using P2 reagents with 61 bp paired-end reads.

## Synaptosome mRNA library preparation and sequencing

Library preparation and sequencing of synaptosomes was performed using a modified version of an established protocol[20,157]. In brief, to lyse membranes and digest DNA, synaptosomes were resuspended in 7.5 μL of 1 mM DTT, 1 U/μL Recombinant RNAse Inhibitor (2313A, Takara Bio), 1 X SingleShot Lysis Buffer and 1 X DNAse Solution (1725080, Bio-rad) and incubated for 20 min at 20 °C, followed by enzyme inactivation for 5 min at 75 °C. Next, reverse transcription with template switching was carried out. We first hybridized indexed RT primers (final concentration: 1 mM) (see Supplementary Data 5), adding DTT (10 mM), dNTPs (1 mM), dCTPs (1.5 mM) and Recombinant RNAse Inhibitor (1 U/μL) and incubating each sample for 5 min at 65 °C and then immediately transferred the samples to ice. After adding the reverse transcriptase SuperScript IV (10 U/μL) with SuperScript RT buffer (1 X; Thermo Fisher Scientific, 18090050), MgCl₂ (6 mM), Betaine (1 M, Merck B0300-5VL) and template switch oligo (1 μM, 5′-/5BiosG/AAGCAGTGGTATCAACGCAGAGTGTCGTGAC TGGGAAAACCCTGGGCrGrGrG-3′) the sample was incubated for 10 min at 55 °C, followed by enzyme inactivation for 10 min at 80 °C. For PCR, 1 X of the DNA polymerase mix (KAPA HiFi HotStart Ready Mix, 07958935001) and the PCR primer (0.1 μM, 5′-AAGCAGTGG-TATCAACGCAGAGT-3′) were added. PCR was then carried out using the following cycling conditions: 3 min incubation at 98 °C for activation of the DNA polymerase, 18 cycles of incubation at 98 °C for 20 s to denature DNA, 67 °C for 10 s for annealing and 72 °C for 6 min for extension. After the last cycle, the sample was again incubated at 72 °C for a final extension phase. Between incubations, samples were constantly kept on ice. After PCR, the Whatman GF/F glass microfiber filter was removed by transferring the sample to a filter column (Spin-X Centrifuge Tube Filter, 0.45 uM in 2.0 ml Tube; Corning, 8162) and centrifuged at 3000 × g for 1 min. Next, 28.8 μL AMPure XP beads (Beckman Coulter) were used for DNA purification. Multiplexed samples were subsequently pooled and further prepared for sequencing using the Nextera XT DNA library prep kit (Illumina) according to the manufacturer's instructions, but using a custom P5

primer (5′-AATGATACGGCGACCACCGAGATCTACACGCCTGTCCGC GGAAGCAGTGGTATCAACGCAGAGTAC-3′). Paired-end sequencing was performed using the NextSeq™ 2000 system (Illumina) with NextSeq™ 1000/2000 P2 (100 cycles) and a custom sequencing primer for Read 1 (5′- GCCTGTCCGCGGAAGCAGTGGTATCAACGCA-GAGTAC- 3′). The following sequencing parameters were used: Read 1 = 26 bp, Read2 = 82 bp, Read 1 Index = 8 bp.

## Sample processing of tissue for LC-MS/MS

Frozen tissue was briefly thawed and mechanically lysed by addition of 100 μL lysis buffer (5% SDS, 50 mM Tris, pH 7.55 with HCl) and 10 'strokes' with a pestle (VWR International) after which the pestle was washed with 100 μL additional lysis buffer. Further homogenization of lysates was then achieved by pipetting up and down 3 times, 10 min sonication and 5 min at 75 °C and 400 rpm. After cooling, lysates were incubated with benzonase (1 μL; 250 units/mL stock solution; Sigma) for 10 min at room temperature. To clear debris, lysates were centrifuged for 10 min at 13,000 × g and the supernatant was taken. Protein concentration of lysates was determined by a BCA assay (Thermo Fisher Scientific) from a 1:5 dilution to ensure minimal interference from the high detergent concentration.

The samples were prepared for bottom-up proteomics using a suspension trapping protocol as previously reported[158]. Briefly, 30 μg of protein in lysis buffer was reduced using 20 mM DTT for 10 min at RT and alkylated with 50 mM iodoacetamide for 30 min at RT in the dark. Afterwards, samples were acidified using phosphoric acid to a final concentration of -1.2%. Binding/wash buffer (BW: 90% methanol, 50 mM Tris, pH 7.1 with H₃PO₄) was added in a 1:7 ratio. The protein suspension was loaded onto the S-trap filter (size: "micro"; ProtiFi) in 150 μL steps by centrifugation for 20 s at 4000 × g and trapped proteins were then washed with 150 μl of BW buffer three times. Digestion buffer was prepared by addition of trypsin (1 μg; Promega) and LysC (0.2 μg; Promega) to 40 μL ammonium bicarbonate buffer buffer (ABC: 50 mM). Digestion was performed overnight (-18 h) at RT in a humidified chamber under gentle agitation. Peptides were later eluted by centrifugation at 4000 × g for 60 s via one wash with 40 μL ABC buffer and two additional washes with 0.2% formic acid in MS-grade water.

After digestion, peptides were desalted using a modified C18 StageTip protocol[159]. StageTips were made by addition of two disks of C18 material (Empore 3M) to a 200 μL pipette tip (Eppendorf). Disks were conditioned using 100 μL pure methanol and centrifuged at 2000 × g for 3 min, after which they were washed with 100 μL 50% acetonitrile with 0.5% acetic acid and equilibrated with two washes of 100 μL 0.5% acetic acid. Peptides were loaded and reloaded on disks by centrifugation at 1500 × g for 5 min and desalted by washing with 100 μL 0.5% acetic acid twice at 2000 × g for 3 min. Desalted peptides were eluted using 75 μL 50% acetonitrile with 0.5% acetic acid and centrifuged at 2000 × g for 3 min, repeating the elution twice. Eluted peptides were dried *in vacuo* at 45 °C.

## Sample processing of synaptosomes for LC-MS/MS

Sorted synaptosomes and control particles were filtered onto Whatman glass microfiber filters (GF/F, Cytiva) as described above and stored at −79 °C in Eppendorf tubes until further processing as previously described[5]. After thawing, digestion buffer consisting of trypsin (0.1 μg) and LysC (0.1 μg) in 25 μL triethylammonium bicarbonate (TEAB; 50 mM) was added to each sample. Digestion was performed overnight (-18 h) at 37 °C. 50 μL pure acetonitrile was then added and samples were centrifuged at 16,000 × g for 10 min. A ZipTip pipette tip (Millipore) was washed once using 50% acetonitrile in MS-grade water, after which supernatants were filtered through by centrifugation for 1 min at 2000 × g. 50 μL 50% acetonitrile was then added to each sample and the process was repeated, pooling both supernatants during filtering. Eluted peptides were dried *in vacuo* at 45 °C.

## LC-MS/MS analysis

Dried peptides were reconstituted in 5% acetonitrile (ACN) with 0.1% FA. For synaptosome samples, reconstitution buffer was supplemented with iRT peptide standard (Biognosys) at 1:100. Reconstituted peptides were loaded onto a C18-PepMap 100 trapping column (particle size 3 μm, $L$ = 20 mm, Thermo Fisher Scientific) and separated on a C18 analytical column with an integrated emitter (particle size = 1.7 μm, ID = 75 μm, $L$ = 50 cm; CoAnn Technologies) using a nano-HPLC (U3000 RSLCnano, Dionex) coupled to a nanoFlex source (2000 V, Thermo Fisher Scientific). Temperature of the column oven was maintained at 55 °C. Trapping was carried out for 6 min with a flow rate of 6 μL/min using loading buffer (98% $H_2O$, 2% ACN with 0.05% TFA). Peptides were separated by a gradient of water (buffer A: 100% $H_2O$ and 0.1% FA) and acetonitrile (buffer B: 80% ACN, 20% $H_2O$ and 0.1% FA) with a constant flow rate of 250 nL/min. In 155 min runs, peptides were eluted by a non-linear gradient with 120 min active gradient time, as selected for the respective MS method by Muntel et al.[160]. Analysis was carried out on a Fusion Lumos mass spectrometer (Thermo Fisher Scientific) operated in positive polarity and data independent acquisition (DIA) mode. In brief, the 40-window DIA method had the following settings: Full scan: orbitrap resolution = 120k, AGC target = 125%, mass range = 350–1650 $m/z$ and maximum injection time = 100 ms. DIA scan: activation type: HCD, HCD collision energy = 27%, orbitrap resolution = 30k, AGC target = 2000%, maximum injection time = dynamic.

## Processing of DIA LC-MS/MS data

DIA raw files were processed with the open-source software DIA-NN (version 1.8.2 beta 27) using a library-free approach. The predicted library was generated using the in silico FASTA digest (Trypsin/P) option with the UniProtKB database (Proteome_ID: UP000000589) for *mus musculus*. Deep learning-based spectra- and RT-prediction was enabled. The covered peptide length range was set to 7-35 amino acids, missed cleavages to 2 and precursor charge range to 1–5. Methionine oxidation was set as variable and cysteine carbamidomethylation as a fixed modification. The maximum number of variable modifications per peptide was limited to 3. According to most of DIA-NN's default settings, MS1 and MS2 accuracies as well as scan-windows were set to 0, isotopologues and match-between-runs were enabled, while shared spectra were disabled. Protein inference was performed using genes with the heuristic protein inference option enabled. The neural network classifier was set to single-pass mode and the quantification strategy was selected as 'QuantUMS (high precision)'. The cross-run normalization was set to 'RT-dependent', library generation to 'smart profiling' and speed and Ram usage to 'optimal results'. The DIA-NN report table and the respective fasta file were imported in the statistical computing software R and analyzed using the MS-DAP R package (v1.2.1)[161].

## Preprocessing methods for RNA-Seq tissue datasets

To process the Illumina NextSeq 2000 sequencing data, we implemented a comprehensive bioinformatics pipeline running on a high-performance computing cluster with allocated resources of one node, 32 CPUs and a runtime limit of 100 h. The pipeline encompasses steps from raw data demultiplexing to the quantification of gene expression levels. (i) *Demultiplexing*. Initially, raw BCL files generated by the Illumina NextSeq 2000 system were converted to FASTQ format using the Illumina bcl2fastq2(v2.20.0.422) software. The conversion process was tailored to our experimental design by providing a custom sample sheet, enabling the direct demultiplexing of samples without lane splitting. This step utilized 8 threads for loading, processing and writing, optimizing throughput. (ii) *Quality Control and Adapter Trimming*. Subsequent to demultiplexing, the FASTQ files underwent quality control and adapter trimming using fastp(0.23.1)[162]. This utility was configured to remove adapter sequences, specified both as

individual sequences and as a FASTA file and to trim low-quality bases and poly-X sequences. The quality threshold was set to a minimum average quality score of 20, with a minimum poly-X length of 10 for trimming. This process was parallelized across 16 threads to expedite execution. Each sample's processing generated a corresponding report in JSON and HTML formats, along with a textual summary. (iii) *Alignment*. The cleaned FASTQ files were aligned to the *Mus musculus* (mm39) reference genome using the STAR (2.7.11b) aligner[163]. The alignment was conducted with the aim of generating sorted BAM files, considering only uniquely mapped reads and filtering out non-canonical intron motifs. Alignment was performed in a strand-specific manner to account for RNA-Seq library preparation nuances. Post-alignment, each sample's BAM file was indexed with samtools for efficient downstream processing. (iv) *Alignment Summary*. An alignment summary was compiled for each sample, detailing the total number of input reads and uniquely mapped reads. This step involved parsing STAR log files and aggregating the relevant metrics into a comprehensive tab-delimited summary file. (v) *Quantification*. For gene expression quantification, featureCounts (v2.0.3)[164] (part of the Subread package[165]) was employed, counting reads mapping to exon regions and annotated with gene IDs. The software was instructed to count paired-end reads as fragments and to include only reads with a mapping quality of 255, ensuring the consideration of high-confidence alignments only. Quantification was carried out on all samples simultaneously, utilizing 16 threads to improve efficiency and results were compiled into a single output file.

## Preprocessing methods for RNA-Seq of synaptosome datasets

Barcoded synaptosome RNA sequencing data was processed using an automated pipeline incorporating demultiplexing, quality filtering and alignment. Raw sequencing data was demultiplexed using 'bcl2fastq'[166], converting base-call files into FASTQ format. Alignment and quantification were performed using 'STAR' (v2.7.11b[163]) in 'STARsolo' mode with '--soloType CB_UMI_Simple', integrating read mapping, barcode error correction ('--soloCBmatchWLtype 1MM'), UMI deduplication ('--soloUMIdedup 1MM_All') and gene expression quantification ('--soloFeatures GeneFull_Ex50pAS'). The data was aligned against the *Mus musculus* (mm39) reference genome (Genome Reference Consortium) and annotation from UCSC Genome Browser. The resulting BAM files, gene expression matrices and alignment logs were systematically organized for downstream analyses. The code to the pipeline is publicly available under: https://gitlab.mpcdf.mpg.de/mpibr/schu/synaptosort.

## Differential expression analysis of transcriptomes

Differential expression analysis was conducted in R 4.4.2 using the DESeq2 (v1.46.0) standard workflow[167]. The design formula included covariates for replicate and region/strata to account for technical/biological variability and spatial origin, respectively. Quality control measures involved assessing read yield versus alignment rate and identifying PCA outliers. Matrices were filtered to include only protein-coding mRNA transcripts from the annotated *Mus musculus* (mm39) reference genome (Genome Reference Consortium). Only replicates passing quality checks were used for the final PCA plots and any replicates failing to meet the quality checks were excluded from downstream analysis (SR replicate 4 and SLM replicate 5 were excluded for failing quality control). For the both tissue and sorted synaptosomes, DESeq2 objects were created separately for regions and strata, with CA1 as the reference for differential expression analysis. For synaptosome datasets, DESeq2 objects were generated for sorted (FMdye-and tdTomato-positive, P3 gate) fraction over (FMdye-positive only, P2 gate) control fraction as a reference to compile a contaminants list with which the data was filtered. Effect size shrinkage was applied using the *apeglm* method (v1.28.0) and statistical significance was assessed using *s values*[167–169], calculated based on the

local false sign rate (lfsr). Differential expression testing used the negative binomial Wald test as implemented in DESeq2 (two-sided). Effect size is represented by the $\log_2$ fold change. Enrichment lists were derived from the differential expression analysis. Significantly enriched 'marker' transcripts were defined as those with a $\log_2$ fold-change 0.1 in the selected region or stratum, meeting statistical significance with an $s$ value of <0.01 for tissue and <0.05 for synaptosomes. For the rank plots, normalized mRNA counts for the tissue subregions and strata were averaged and log-transformed for ranking of genes. $Z$-scores of $\log_2$ fold changes used in heatmaps were computed using the scale function in pheatmap (v1.0.13)[170]. For ridge plots, $z$-scores of fold changes were manually calculated by subtracting the mean fold change across all strata from the fold change in a given strata and dividing by the standard deviation. Transcripts were included in each ridge plot on the basis of their gene assignment to GO terms potassium channel complex (GO:0034705), sodium channel complex (GO:0034706), calcium channel complex (GO:0034704), GPCR activity (GO:0004930), and mitochondrion (GO:0005739). Partial Least Squares Discriminant Analysis (PLS-DA) was performed to classify synaptosome subregions and strata. A threefold cross-validation with 15 repetitions was implemented to optimize model performance. The model was trained using the *pls* method (v2.8.5) in the *caret* package (v7.0.1)[171], with oneSE as the optimal tuning parameter and pre-processing steps including zero variance removal, centering and scaling. The top 10 (subregions) or top 5 (strata) loadings from each of the first three components were selected and their VIP (Variable Importance in Projection) scores plotted. $Z$-scores of $\log_2$ fold changes shown in heatmaps were computed using the scale function in pheatmap[170]. Data wrangling and visualization were performed using dplyr (v1.1.4), stringr (v1.5.1), ggplot2 (v3.5.2)[172] and heatmaps were generated with pheatmap (v1.0.13)[170].

### Data analysis of proteomes

Tissue data were analyzed using MS-DAP (v1.2.1) in the R environment with the following parameters: filter minimum detect = 1, filter minimum quant = 1, filter minimum peptide per protein = 1, filter by contrast = true, normalization algorithm = rlr. MSqRob was selected as the differential expression algorithm with a $q$-value threshold of 0.01 and a $\log_2$ fold change cutoff of 0.1. PCA was performed using $\log_2$ intensities of all quantified protein groups. $Z$-scores of $\log_2$ fold changes used in heatmaps were computed using the scale function in pheatmap[170]. For ridge plots, $z$-scores of fold changes were manually calculated by subtracting the mean fold change across all strata from the fold change in a given strata and dividing by the standard deviation. Proteins were included in each ridge plot on the basis of their gene assignment to GO terms potassium channel complex (GO:0034705), sodium channel complex (GO:0034706), calcium channel complex (GO:0034704), GPCR activity (GO:0004930), and mitochondrion (GO:0005739).

Synaptosome data were analyzed using MS-DAP with the same quantification and normalization parameters as tissue. For differential expression, the MS-EmpiRe algorithm was used to minimize shrinkage of very small fold change differences between sample groups. The $q$-value threshold was set at 0.05 and $\log_2$ fold change threshold at 0.1. An initial analysis was done comparing sorted (FMdye- and tdTomato-positive) fractions to unsorted (FMdye-positive only) fractions for each subregion/strata. Proteins enriched in the unsorted fraction for any subregion or any strata were compiled to form a list of subregion and strata "contaminant" proteins that were filtered out of the dataset prior to subsequent analysis. To then determine subregion-enriched synaptic proteins, the sorted fractions were compared directly for each subregion. To determine strata-wise enrichment, a comparison was performed between the sorted fractions from each strata and a background consisting of all remaining strata. For classification of subregion and strata synaptic proteomes using PLS-DA, only proteins with no missing values across all subregion or strata samples were included. Threefold cross validation was used for both datasets with 15 repetitions for subregions and 100 repetitions for strata. Model training was done with the *pls* method in the *caret* package[171], with oneSE as the optimal tuning parameter and pre-processing steps of zero variance removal, centering, and scaling. $Z$-scores of $\log_2$ fold changes shown in heatmaps were computed using the scale function in pheatmap[170]. Plots were visualized using ggplot2 (v3.5.1)[172]. Heatmaps were visualized using pheatmap (v1.0.12)[170].

### Functional enrichment analysis of individual transcriptomes and proteomes

GO overrepresentation analysis of both transcriptome and proteome tissue data was performed using the ClusterProfiler R package (v4.12.6) (https://github.com/YuLab-SMU/clusterProfiler). An annotation file was fetched for *Mus musculus* from OrgDB (v3.19.0) using the access code AH116711. Gene lists of significantly enriched proteins or transcripts in each strata determined by previous differential expression analysis were compared to a custom background consisting of all genes identified in the respective experiment. Analysis was carried out with a Fisher's exact test using gene symbols, the ontology setting "all" (encompassing biological processes, cellular compartment and molecular function), Benjamini-Hochberg FDR correction and an FDR cut-off of 0.05. Redundant GO terms were simplified using DOSE (v4.0.1) according to the adjusted $p$-value with a cut-off of 0.7 using the measure Wang. Enrichment results were then merged by count for visualization of the top-7 terms per condition.

### Integration of transcriptomic and proteomic data sets

All comparisons between protein and mRNA data were done using $\log_2$ scaled DESeq2 normalized counts and intensity Based Absolute Quantification (iBAQ) values. iBAQ values were determined using the DIAgui shiny app (v1.4.5)[173], quantifying based on unique and shared peptides and filtering for precursor $q$-value < 0.01 and protein-group $q$-value < 0.01. Additional parameters included Use MaxLFQ from diann package, only keep peptide counts all, and peptide length 7-35.

For direct 1:1 analysis of the population level relationship between transcripts and their corresponding protein, Pearson's correlations were calculated for complete observations of transcripts and proteins using their mean abundance in each strata and across all strata. To analyze the correlation of variation in individual transcript and protein abundances between strata, row-wise Pearson's correlation was performed on mean values for each strata. Transcript-protein pairs were then grouped based on strong positive ($r \geq 0.9$) or negative ($r \leq -0.9$) correlation. GO overrepresentation analysis of these groups was performed as described above, with all detected mRNA transcripts used as the background. Integration with half-life data was performed by merging with a half-life dataset from Dörrbaum et al.[118] based on gene names. To compare protein half-lives across gene subsets, normality was assessed via Q-Q plots. As the data were not normally distributed, a non-parametric Kruskal–Wallis test was applied followed by a post hoc Dunn's test with Bonferroni correction. No data were excluded, and no assumptions were made about equal variances. For co-enrichment analysis, genes were grouped into categories based on which strata the mRNA and protein showed significant enrichment in. Translation localization was determined by gene-wise merging with Glock, Biever et al.[19] translatome dataset showing significant translation differences between somatic- and neuropil-enriched CA1 areas. All analyses were performed in R.

**Immunofluorescence staining in hippocampal slices.** Anesthesia and euthanasia procedures for mice used in immunofluorescence staining were performed as previously described in the 'Animals used for transcriptomics and proteomics' section above: briefly, mice were decapitated following isoflurane anesthesia. A previous protocol was adapted for mouse tissue[19]. Brains of 10/11-week-old mice were fixed

with 4% (volume/volume) paraformaldehyde (PFA) in 4% (weight/volume) sucrose dissolved in 1X phosphate buffered saline (PBS) for 2 h at room temperature, washed thrice with 1X PBS and then incubated in 15% sucrose in PBS overnight at 4 °C. The following day, the solution was changed and the brains were incubated in 30% sucrose in 1X PBS for 72 h at 4 °C. Hippocampi were cryosectioned at 40 μm thickness the same way as previously described[82] and transferred to a 24-well plate with 1X PBS and washed twice with 1X PBS. The slices were permeabilized overnight in 0.5 mL 0.5% Triton-X-100 in blocking buffer (4% goat serum in 1X PBS) at 4 °C with rocking. The following morning, the slices were blocked with 1 mL blocking buffer for 6–8 h at 4 °C with rocking. The slices were then incubated with 0.8 mL primary antibodies (see Supplementary Data 6) diluted in blocking buffer overnight at 4 °C with rocking. The following morning, the slices were washed thrice with 1X PBS and incubated with DAPI (1:1000) and secondary antibodies (Rabbit-anti-Mbp, Thermo Fisher, #PA5-78397, 1:500; Rabbit-anti-Hnrpdl, Proteintech, #10660-2-AP, 1:800; Rabbit-anti-Lrrtm1, Alomone Labs, #ANR-141, 1:500; Rabbit-anti-mGlur2, Thermo Fisher, #MA5-42460, 1:200; Rabbit-anti-Homer2, SYSY, #160 203, 1:500; Rat-anti-Nectin3, Abcam, #Ab16913, 1:100; Rabbit-anti-Calb2, Abcam, #Ab16694, 1:100; Alexa Fluor 488 goat-anti-rabbit, Thermo Fisher, #A11008, 1:1000; Alexa Fluor 488 goat-anti-rat, Thermo Fisher, #A11006, 1:1000, see Supplementary Data 6) diluted in 0.8 mL blocking buffer for 4–5 h at room temperature with rocking. After three washes with 1X PBS, the slices were mounted on glass slides (Epredia J1800AMNZ) with Aqua Poly/mount (Polysciences 18606).

## Microscopy

Samples were imaged using either the Zeiss LSM780 or the Zeiss LSM980 confocal microscope using the Plan-Apochromat 20x/0.8 M27 objective. Z-stacks of 15–20 μm at an interval of 1–2.47 μm were used to cover the range of antibody signals, and 5 by 5 tile scans of 512/512 pixel tiles were stitched together to produce complete images. Laser intensity and detector gain were set to avoid saturated pixels. For visualization purposes, maximum intensity projections were adjusted for brightness and contrast using ImageJ analysis software[174].

## Reporting summary

Further information on research design is available in the Nature Portfolio Reporting Summary linked to this article.

## Data availability

The raw RNA-sequencing data for tissue and synaptosomes are available at the NCBI Sequence Read Archive (SRA)[175,176] under the reference number: PRJNA1142450 ("RNAseq of microdissected mouse hippocampal subregions and CA1 strata", [https://dataview.ncbi.nlm.nih.gov/object/PRJNA1142450]) and PRJNA1142761 ("RNA-Seq of sorted synaptosomes from microdissection of mouse hippocampal subregions and CA1 strata", [https://dataview.ncbi.nlm.nih.gov/object/PRJNA1142761]). Mass spectrometry data associated with this manuscript have been uploaded to the PRIDE[177] repository (Project accession: PXD054879, [https://www.ebi.ac.uk/pride/archive/projects/PXD054879]). mRNA and proteins can also be assessed at our lab's public interface https://syndive.org/. Source data are provided with this paper.

## Code availability

All customized R scripts are deposited on Gitlab: https://gitlab.mpcdf.mpg.de/mpibr/schu/HippocampusMicrodissections or under the following https://doi.org/10.17617/1.1WVD-V337

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

## Acknowledgements

We thank Susanne tom Dieck, Marc van Oostrum and the rest of the Schuman Lab and proteomics facility for technical support and assistance. The MPIBR support facilities, in particular Silke Zeissler and her team from Animal facility, Alla Schleif from media and glassware, Christine Bohnstaedt from the Imaging facility, the team from Scientific computing, and Julia Kuhl for graphical support. R.R. was supported through a Fulbright grant of the German-American Fulbright Commission. E.K. acknowledges funding by EMBO (Postdoctoral Fellowship EMBO ALTF 148-2023). The lab of E.M.S. is funded by the Max Planck Society and the European Union (ERC, DiverseSynapse, 101054512). Funded by the European Union. Views and opinions expressed are, however, those of the author(s) only and do not necessarily reflect those of the EU or the ERC. Neither the EU nor the granting authority can be held responsible for them.

## Author contributions

E.K. co-wrote the manuscript, performed microdissections, FASS sorting, RNA library preparation, RNA-sequencing, transcriptomic- and multi-omic analyses. Q.W. co-wrote the manuscript and carried out proteomic sample preparation, LC-MS/MS, proteomic- and multi-omic analyses. N.F. performed microdissections and synaptosome preparations. K.D. contributed to proteomic analysis. J.M. performed preprocessing of transcriptomic data. E.C. contributed to tissue library preparation and sequencing. M.J. developed synaptosome library preparation protocol. R.R. and B.N-A. carried out proteomics validations. G.T. contributed to transcriptomic analysis. J.L. supervised proteomics. E.M.S. supervised the overall project and co-wrote the manuscript.

## Funding

## Competing interests

The authors declare no competing interests.
