## [Transparent Peer Review file · Nature Communications]

An integrated transcriptomic and proteomic map of the mouse hippocampus at synaptic resolution

Corresponding Author: Dr Erin Schuman

Version 1:

Reviewer comments:

Reviewer #1

(Remarks to the Author)

Resubmission of Kaulin & Waselenchuk et al., presents a highly revised and adjusted manuscript, which takes into account the extensive comments provided from the last round of revision. Direct comparisons between the two submissions is challenging given the extent of changes, however the authors have successfully addressed key concerns previously raised, as well as highlighted key caveats where necessary, and as a result the manuscript has been improved. Furthermore, they have succinctly addressed concerns around comparing whole tissue vs. purified synaptosomes to draw conclusions based on highly differing methods for measurement. Finally, the newly analysed data clearly comparing gene and protein expression across subregions is highly valuable and presented clearly.

Authors now clearly show which genes and proteins are significantly enriched in each subregion and synaptosomes. This is exemplified by the improved Figures 1G, 2F, 3B, 3C, 3D, 6E and 7E. Supplemental Tables 1 & 4 contains genes which are significantly enriched in a given region. This greatly improves accessibility of the data. However, authors should include gene names for not significantly different genes to improve data completeness and consistency.

The authors have used published RNA-ISH images from the Allen Brain Atlas to confirm their transcription data, as well as IF analysis to confirm their region-specific proteomics (Extended Data S3 and S4). While these largely do correspond to their global analysis, Map2 in S4 is found enriched in the SR by transcriptomics, but by RNA-ISH is clearly enriched in the SP. Can authors suggest why? This could indicate contamination in the dissections, and question which other transcripts have been falsely annotated.

Figure 6D the PCA axes are missing the % variance, this needs to be added.

Aside from the points raised above I have no further comments and congratulate the authors on producing such an improved manuscript which I am sure will be of use to the community.

(Remarks on code availability)

Reviewer #2

(Remarks to the Author)

The authors have addressed many of my concerns; however, some points remain insufficiently resolved. PLS-DA is a supervised method that clusters samples based on predefined groups and, therefore, cannot replace PCA for unbiased visualization of sample clustering. As such, data variability remains an underlying concern that has not been adequately addressed. Additionally, the authors state that the quality of their synaptosome workflow has been extensively validated previously (van Oostrum et al.; PMID: 37918396). However, this does not address my question regarding the independent validation of RNA and protein localization at synapses, particularly since their study involves different brain regions where

artifacts may differ.

(Remarks on code availability)

I cannot easily get access to the code. The provided link (<https://gitlab.mpcdf.mpg.de/mpibr/schu/HippocampusMicrodissections/>) requires sign-in in and new accounts require an MPCDF invitation.

Reviewer #3

(Remarks to the Author)

The authors have done an excellent job responding to my comments and critiques. The figures and manuscript are significantly improved.

I fully support publishing in NC.

Jeff Savas

(Remarks on code availability)

REVIEWER COMMENTS

Reviewer #1 (Remarks to the Author):

Resubmission of Kaulin & Waselenchuk et al., presents a highly revised and adjusted manuscript, which takes into account the extensive comments provided from the last round of revision. Direct comparisons between the two submissions is challenging given the extent of changes, however the authors have successfully addressed key concerns previously raised, as well as highlighted key caveats where necessary, and as a result the manuscript has been improved. Furthermore, they have succinctly addressed concerns around comparing whole tissue vs. purified synaptosomes to draw conclusions based on highly differing methods for measurement. Finally, the newly analysed data clearly comparing gene and protein expression across subregions is highly valuable and presented clearly.

Authors now clearly show which genes and proteins are significantly enriched in each subregion and synaptosomes. This is exemplified by the improved Figures 1G, 2F, 3B, 3C, 3D, 6E and 7E. Supplemental Tables 1 & 4 contains genes which are significantly enriched in a given region. This greatly improves accessibility of the data. However, authors should include gene names for not significantly different genes to improve data completeness and consistency.

Regarding the inclusion of non-significantly different genes: The complete datasets are available in Supplemental Tables 1 and 4, ensuring data completeness and consistency. For the transcriptomics data, transcripts that are not significantly enriched are explicitly labeled as "NO" in the "diffexpressed" column to indicate lack of significant differential expression (UP or DOWN indicates that they are significantly enriched or depleted, respectively). Similarly, for the proteomics data, all proteins are included, with the significance indicated by "TRUE" for significantly regulated proteins and "FALSE" for those not significantly different.

The authors have used published RNA-ISH images from the Allen Brain Atlas to confirm their transcription data, as well as IF analysis to confirm their region-specific proteomics (Extended Data S3 and S4). While these largely do correspond to their global analysis, Map2 in S4 is found enriched in the SR by transcriptomics, but by RNA-ISH is clearly enriched in the SP. Can authors suggest why? This could indicate contamination in the dissections, and question which other transcripts have been falsely annotated.

We acknowledge the apparent discrepancy between the transcriptomic enrichment of Map2 in the SR and its RNA-ISH localization predominantly in the SP, as shown in the Allen Brain Atlas. However, as the cell body is the site of transcription, every mRNA will be present there in high abundances - this is evident in other ISH images as well where SP generally appears more densely populated with mRNAs than any

other area. This is taken into account during our analysis and normalization such that enrichment values reflect relative differences, taking into account the abundances of all other mRNAs in the region/strata as well. Moreover, previous studies have shown enrichment of *Map2* in SR (Garner et al., 1988, DOI: 10.1038/336674a0), and work from our lab has also shown that *Map2* is preferentially translated in hippocampal neurites compared to somata (Glock, Biever et al., 2021, PMID: 34670838).

Figure 6D the PCA axes are missing the % variance, this needs to be added. Figure 6 shows PLS plots, not PCA, and % variance is accordingly not shown.

Aside from the points raised above I have no further comments and congratulate the authors on producing such an improved manuscript which I am sure will be of use to the community.

Reviewer #2 (Remarks to the Author):

The authors have addressed many of my concerns; however, some points remain insufficiently resolved. PLS-DA is a supervised method that clusters samples based on predefined groups and, therefore, cannot replace PCA for unbiased visualization of sample clustering. As such, data variability remains an underlying concern that has not been adequately addressed. Additionally, the authors state that the quality of their synaptosome workflow has been extensively validated previously (van Oostrum et al.; PMID: 37918396). However, this does not address my question regarding the independent validation of RNA and protein localization at synapses, particularly since their study involves different brain regions where artifacts may differ.

We acknowledge that PLS-DA is a supervised method that maximises covariance between different classes and provides different information compared to PCA which aims to maximize variance. PLS-DA is better suited for classification tasks where the goal is to separate different groups based on important features. Our aim was to classify synapses from one neuronal population based on key molecules that are synapse-specific molecular signatures. PLS-DA was therefore chosen to highlight strata-specific synaptic features relevant to this question. We note that for the analysis associated with hippocampal tissue, we did use PCA (Figure 1d, Figure 2d).

Regarding data variability, we have now included scatter plots for synaptosome data illustrating that samples within groups correlate with one another highly (now included as Extended Data Figure S6 & S7). We recognize that for the proteome, SP replicate one shows slightly lower correlation values (~0.8). However, we did not deem this sufficient to exclude it from subsequent analysis, as we prioritized keeping replicates to ensure robust statistics could be performed.

We have updated the figure reference from S6 to S8 throughout the main manuscript, including in Lines 370 and 380. Additionally, we modified the sentence in Lines 358–363 by adding the following bolded text: “After filtering for residual contaminant molecules enriched in a precursor fraction, the number of quantified proteins and detected mRNA transcripts across subregion and strata synapses totalled > 5,000 and 15,000, respectively, **with high reproducibility between replicates** (Fig. 6c, Fig. 7b, **Extended Data Figs. S6-S7**, Extended Data Table S2 and S4).

Regarding synaptic localization, our synaptosome isolation and purification (FASS) workflow is identical to the extensively validated and in house developed workflow outlined in van Oostrum et al., 2023 (PMID: 37918396) and Hafner, Donlin-Asp et al., 2019 (PMID: 31097639). van Oostrum et al. also similarly includes synaptosomes from subregions (CA1, CA3, DG) of the hippocampus. Importantly, we make use of 2 fractions: a less pure “control” (P2) fraction, and a highly pure P3 fraction. The lower purity of the control fraction means that we can assume that all molecules *enriched* here are not synaptically localized. This was incorporated into our analysis (as stated in the methods), and all P2 enriched molecules have been excluded. Strongly P2 enriched molecules included myelin and glial markers, illustrating that this serves as an effective control. This comparison was previously suggested to be highly stringent.

Reviewer #2 (Remarks on code availability):

I cannot easily get access to the code. The provided link (<https://gitlab.mpcdf.mpg.de/mpibr/schu/HippocampusMicrodissections/>) requires sign-in in and new accounts require an MPCDF invitation.

The complete code was initially included as a PDF file alongside the manuscript. However, we have now updated the repository settings to ensure that the code is fully accessible via the provided link: <https://gitlab.mpcdf.mpg.de/mpibr/schu/HippocampusMicrodissections/> or the DOI: <https://doi.org/10.17617/1.1WVD-V337>. The repository is set to public, so the code can be viewed and cloned without requiring sign-in or an MPCDF account. Please note that an account is still required only for actions such as opening issues or contributing changes. Everything should now be accessible.

Reviewer #3 (Remarks to the Author):

The authors have done an excellent job responding to my comments and critiques.
The figures and manuscript are significantly improved.

I fully support publishing in NC.

Jeff Savas